# A genetic-epigenetic interplay at 1q21.1 locus underlies *CHD1L*-mediated vulnerability to primary progressive multiple sclerosis

Majid Pahlevan Kakhki [1], Antonino Giordano[1,2,3,4,22], Chiara Starvaggi Cucuzza [1,5,22], Tejaswi Venkata S. Badam[1,6,22], Samudyata Samudyata[7,22], Marianne Victoria Lemée [8,9,10,11,22], Pernilla Stridh [1], Asimenia Gkogka[7], Klementy Shchetynsky[1], Adil Harroud [12,13,14], Alexandra Gyllenberg[1], Yun Liu[15], Sanjaykumar Boddul[16], Tojo James[1], Melissa Sorosina[3], Massimo Filippi [2,4,17,18], Federica Esposito[2,3], Fredrik Wermeling [16], Mika Gustafsson [6], Patrizia Casaccia [19], Jan Hillert[1], Tomas Olsson[1], Ingrid Kockum [1], Carl M. Sellgren [7,20,21], Christelle Golzio[8,9,10,11], Lara Kular [1,23] ✉ & Maja Jagodic [1,23] ✉

Multiple Sclerosis (MS) is a heterogeneous inflammatory and neurodegenerative disease with an unpredictable course towards progressive disability. Treating progressive MS is challenging due to limited insights into the underlying mechanisms. We examined the molecular changes associated with primary progressive MS (PPMS) using a cross-tissue (blood and post-mortem brain) and multilayered data (genetic, epigenetic, transcriptomic) from independent cohorts. In PPMS, we found hypermethylation of the 1q21.1 locus, controlled by PPMS-specific genetic variations and influencing the expression of proximal genes (*CHD1L*, *PRKAB2*) in the brain. Evidence from reporter assay and CRISPR/dCas9 experiments supports a causal link between methylation and expression and correlation network analysis further implicates these genes in PPMS brain processes. Knock-down of *CHD1L* in human iPSC-derived neurons and knock-out of *chd1l* in zebrafish led to developmental and functional deficits of neurons. Thus, several lines of evidence suggest a distinct genetic-epigenetic-transcriptional interplay in the 1q21.1 locus potentially contributing to PPMS pathogenesis.

Multiple Sclerosis (MS) is an inflammatory and neurodegenerative disease of the central nervous system (CNS) affecting young adults and leading to unpredictable and progressive physical and cognitive impairments. The pathological hallmarks of MS are represented by focal plaques of primary demyelination and diffuse neurodegeneration in the gray and white matter of the brain and spinal cord[1]. The most common form of disease, relapsing-remitting MS (RRMS), presents itself with neurological relapses followed by periods of partial or complete remission. In most cases, this inflammatory stage is followed by a secondary progressive phase of MS (SPMS). About 10–15% of patients, manifest a primary progressive form of MS disease (PPMS) with uninterrupted progression starting from disease onset, although relapses may occur[2]. Beyond epidemiological differences, both progressive forms of MS share similar features that differ from the RRMS stage, notably, a later onset (40 years vs. 30 years in RRMS) and a prevalence of CNS-intrinsic inflammatory and degenerative pathological processes as opposed to the predominant role of peripheral immunity in the RRMS stage[3]. However, while considerable progress

---

has been achieved in deciphering the genetic variation predisposing for MS, the mechanisms underpinning disease progression and severity remain unresolved. Importantly, conventional immunomodulatory therapies are ineffective in progressive patients[4,5], reinforcing the need to better clarify the underlying processes.

Disease trajectory likely relies on a complex interplay between genetic and non-genetic factors[6]. Epigenetic modifications such as DNA methylation, intersecting internal and external influences, might provide an opportunity to study the mechanisms underlying the different forms of MS disease. DNA methylation, which regulates gene expression without altering the genetic code, is the most studied epigenetic mark and its implications in MS pathogenesis have been reported in immune and nervous cells of MS patients[7–12]. We have shown that the major risk variant *HLA-DRB1*15:01* may mediate risk for MS via changes in *HLA-DRB1* DNA methylation and subsequent expression[11]. Recent comparative whole-genome DNA methylation studies shed further light on molecular mechanisms behind MS pathogenesis[9,13], the vast majority of them focusing on the RRMS stage of the disease.

In this study, we aimed to examine the molecular changes that are associated with MS courses. Using a multilayered (genetic, epigenetic, transcriptomic) and cross-tissue (blood, brain) approach in cohorts of MS patients and controls, we demonstrate an interplay between regional genetic variation, DNA methylation, and gene expression in the chromosome 1q21.1 locus in PPMS patients specifically. We address the link between these three molecular layers using in vitro reporter gene assays and CRISPR/dCas9 epigenome editing. Functional perturbation of *CHD1L* in human induced pluripotent stem cell (iPSC)-derived neuronal and zebrafish models reveal that *CHD1L* knock-down results in developmental and functional impairments in neuronal lineage. Our findings suggest that genetically controlled methylation at 1q21.1 locus might contribute to PPMS pathology through *CHD1L*-modulated structural and functional changes in neurons.

## Results

### Hypermethylation of a region at 1q21.1 in PPMS patients

We profiled genome-wide DNA methylation in the blood of RRMS, SPMS, and rare PPMS patients and matched healthy controls (HC, cohort 1, n = 279, Fig. 1, Supplementary Data 1), using the Illumina Infinium HumanMethylation450 BeadChip (450K array). After adjustment for confounders, we identified one differentially methylated region (DMR) in PPMS patients in comparison with HC ($P = 6.16 \times 10^{-6}$), RRMS ($P = 2.51 \times 10^{-6}$) and SPMS ($P = 7.33 \times 10^{-6}$) patients (FWER < 0.05, Fig. 2a, b, Supplementary Data 2). The DMR, located in an intergenic CG-rich region on chromosome 1 (q21.1): 146549909–146551201 (GRCh37/hg19), consists of 8 consecutive CpG probes displaying hypermethylation in PPMS patients compared to RRMS, SPMS as well as HC, i.e. with a mean [min-max] Δβ of 0.24 [0.02–0.42], 0.23 [0.03–0.40], and 0.24 [0.02–0.43], respectively (Fig. 2b). Considering the low number of PPMS patients (n = 4), we selected an independent cohort for replication including 36 PPMS and 48 RRMS patients, matched for gender, age and disease duration (cohort 2, Fig. 1, Supplementary Data 1). We designed pyrosequencing assays covering seven CpG sites (CpG 1–7), including the 450K probes exhibiting the largest changes cg21263664 (CpG 1), cg03526459 (CpG 3), cg16814483 (CpG 5), cg02487331 (CpG 7) and three additional adjacent CpG sites located between theses 450K-CpGs (Fig. 2a). A total of six out of these seven CpGs were found significantly hypermethylated in PPMS compared to RRMS patients (Fig. 2c). These data imply that PPMS patients display hypermethylation of an intergenic region within the 1q21.1 locus, suggesting a potential involvement in the PPMS pathogenesis.

### Genetic control of 1q21.1 methylation in the blood and CNS

Previous studies have revealed a complex interplay between genetic and epigenetic variations by reporting that part of the epigenome is under genetic control in general[14,15], and more particularly, that genetically dependent DNA methylation can mediate the risk of developing MS[11]. Interestingly, our data revealed that methylation clustered within three methylation levels (i.e., low, medium, and high) in both cohorts (Fig. 2b, c), a pattern typically observed for genetically controlled methylation. To test for potential genetic dependence, we performed genome-wide methylation Quantitative Trait Loci (meQTL) analysis in cohort 1 and found 19 SNPs, located in the extended 1q21.1 locus (−50 kb to +400 kb), associating with methylation levels of CpGs in the DMR, with strongest effects coming from rs1969869, rs21327, rs647596, rs10900384 and rs12401360 (genotype *vs.* methylation estimate = 0.19, $p < 10^{-16}$) (Fig. 3a, Supplementary Data 3). To further validate this finding, we conducted locus-specific meQTL analysis in cohort 2 by testing all SNPs covering ±500 kb window encompassing the DMR against each pyrosequenced CpG. Out of the 123 SNPs tested in cohort 2, two variants showed significant (rs1969869, rs4950357, $p < 10^{-8}$) and two suggestive (rs647596, rs10900384, $p < 10^{-5}$) associations with CpGs methylation (Fig. 3a). An example of genetic control of methylation at CpG 4 (pyrosequenced CpG located between cg03526459 and cg16814483, Fig. 2c) is illustrated in Fig. 3b. The two strongest SNPs, rs1969869 and rs4950357, displayed positive association (β estimates ranging from 1.20 to 1.39) between the minor-allele and CpG methylation levels (Fig. 3a, Supplementary Data 4). In addition to these four SNPs, 66 SNPs demonstrated nominal association with methylation (p < 0.05) (Supplementary Data 4). Overall, all eight overlapping SNPs identified in cohort 1 displayed significant and same direction of changes also in cohort 2 (Fig. 3a, Supplementary Data 3 and 4). Many of the SNPs with similar effects, i.e., minor allele associating with either high or low methylation, were in strong linkage disequilibrium (LD, R2 > 0.8) and were therefore grouped into clusters (Fig. 3a).

Our results, implying a link between genetic variation and blood DNA methylation at the 1q21.1 region, a locus reported to be crucial for brain size[16–18] and neuropsychiatric disorders[19–21], posed the question of whether such an effect might occur in the CNS as well. To address this, we examined putative genetic influences on the methylation at 1q21.1 DMR in the brain using publicly available meQTL data from the brain tissue of 543 individuals[15]. Results showed that SNPs corresponding to 11 out of 14 SNP clusters demonstrated strong meQTLs in the CNS as well, with same direction of the effect as observed in the blood (Fig. 3c, Supplementary Data 5), exemplified for cg02487331 (Fig. 3c). Altogether, these findings strongly suggest that methylation at CpGs in the 1q21.1 DMR are under genetic control in the blood and CNS compartments.

### 1q21.1 methylation-controlling variants predispose for PPMS

We next asked whether the genetic variation in 1q21.1 also affects the risk of developing PPMS. We tested the association between all the imputed SNPs in the extended locus (3057 SNPs, chr.1: 145695987–147526765, hg19) and disease course, comparing PPMS and bout-onset MS (BOMS including RRMS and SPMS) patients in a combined Swedish cohort of 9850 patients (PPMS n = 603, BOMS n = 9247) (cohort 3, Fig. 1, Supplementary Data 1). A total of 574 SNPs, mapping to 57 LD blocks, showed evidence for nominal association (p < 0.05) with PPMS (Supplementary Data 6). Of note, 13 SNPs belonging to seven different LD blocks remained significant after Bonferroni correction for multiple testing ($p < 7.2 \times 10^{-4}$) (Supplementary Data 6). Brain-meQTL data were available for 149/574 SNPs, belonging to 13/57 LD blocks. The vast majority of them (147/149) showed evidence for meQTLs in the brain. All the protective variants were found to be associated with lower methylation (β estimates ranging from −0.08 to −0.34) at 1q21.1 DMR, while 7/8 risk variants for PPMS were associated with higher levels of methylation (β estimates ranging from 0.09 to 0.73) at 1q21.1 DMR in the brain (Supplementary Data 7).

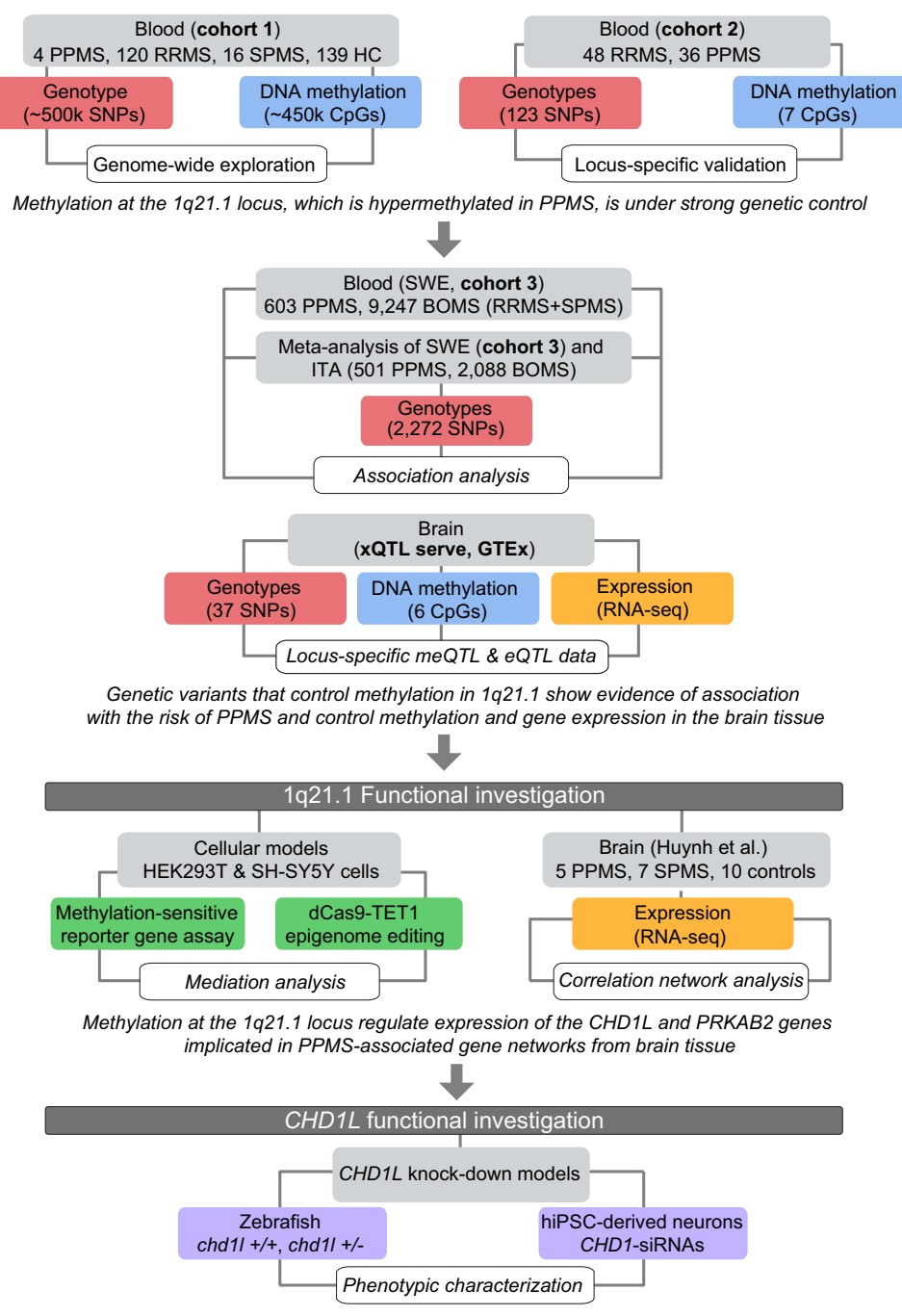

**Fig. 1 | Workflow of the study.** MS multiple sclerosis, PPMS primary progressive MS, SPMS secondary progressive MS, RRMS relapsing-remitting MS, SNP single nucleotide polymorphism, meQTL and eQTL methylation and expression quantitative trait loci, SWE Swedish cohort, ITA Italian cohort, META meta-analysis of the SWE and ITA cohorts.

To further reinforce the evidence of a genetic influence at 1q21.1 on the risk of developing PPMS, we conducted a meta-analysis between the aforementioned Swedish cohort and an Italian cohort of PPMS ($n = 501$) and BOMS ($n = 2088$) individuals. A total of 64 SNPs, from 16 different LD blocks, showed a nominal association ($p < 0.05$) and same direction of effect in both cohorts (Table 1, Supplementary Data 8). Five SNPs, tagging two LD blocks, were associated with an increased risk of PPMS (OR ranging from 1.48 to 1.84) and remained significant after correction for multiple testing accounting for LD blocks ($p < 7.2 \times 10^{-4}$) (Table 1). Annotation to brain-meQTL data

confirmed that risk variants, such as LD block 5 and 6, also lead to higher methylation (β estimates ranging from 0.09 to 0.19, Fig. 3d, Supplementary Data 7). These findings suggest a functional link between high methylation at 1q21.1 DMR and the genetic predisposition to develop PPMS.

## 1q21.1 methylation and PPMS risk variants control expression

We next assessed whether the identified genetic variations that are associated with PPMS and methylation, impact gene expression in the brain tissue. Taking advantage of the GTEx and xQTL serve

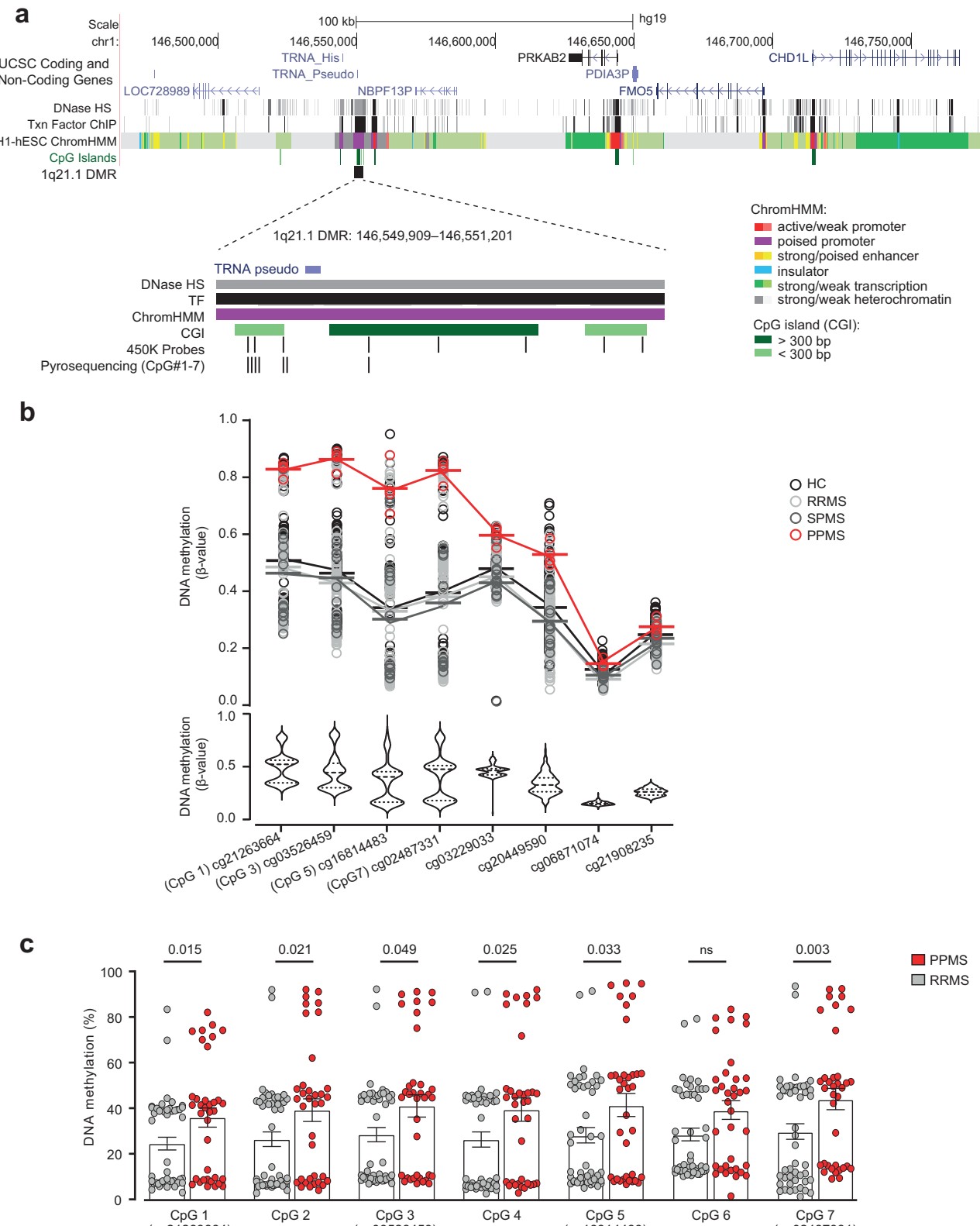

**Fig. 2 | Hypermethylation of the 1q21.1 locus in primary progressive multiple sclerosis patients. a** Genomic annotation of the identified differentially methylated region (DMR) in 1q21.1, including 450K and pyrosequenced CpG probes and regulatory features, i.e. CpG island (CGI) and chromatin state segmentation by hidden Markov model (ChromHMM) from ENCODE/Broad. **b** Dot plot and violin plot of DNA methylation differences at the identified DMR on chromosome 1: 146549909–146551201 (hg19) in PPMS (n = 4) compared to RRMS (n = 120), SPMS (n = 16), and HC (n = 139), obtained using Illumina 450K in cohort 1. **c** Replication of methylation (mean ± SEM) at 7 CpGs (CpG 1–7, including four 450 K probes) in an independent cohort (n_RRMS = 48, n_PPMS = 36, non-parametric two-sided Mann–Whitney U test) using pyrosequencing. MS multiple sclerosis, PPMS primary progressive MS, RRMS relapsing-remitting MS, SPMS secondary progressive MS, HC healthy controls, BOMS bout-onset MS, hiPSC human induced pluripotent stem cells, DNase HS DNase I hypersensitive sites, Txn factor ChIP transcription factor chromatin immunoprecipitation. Source data are provided as a Source Data file.

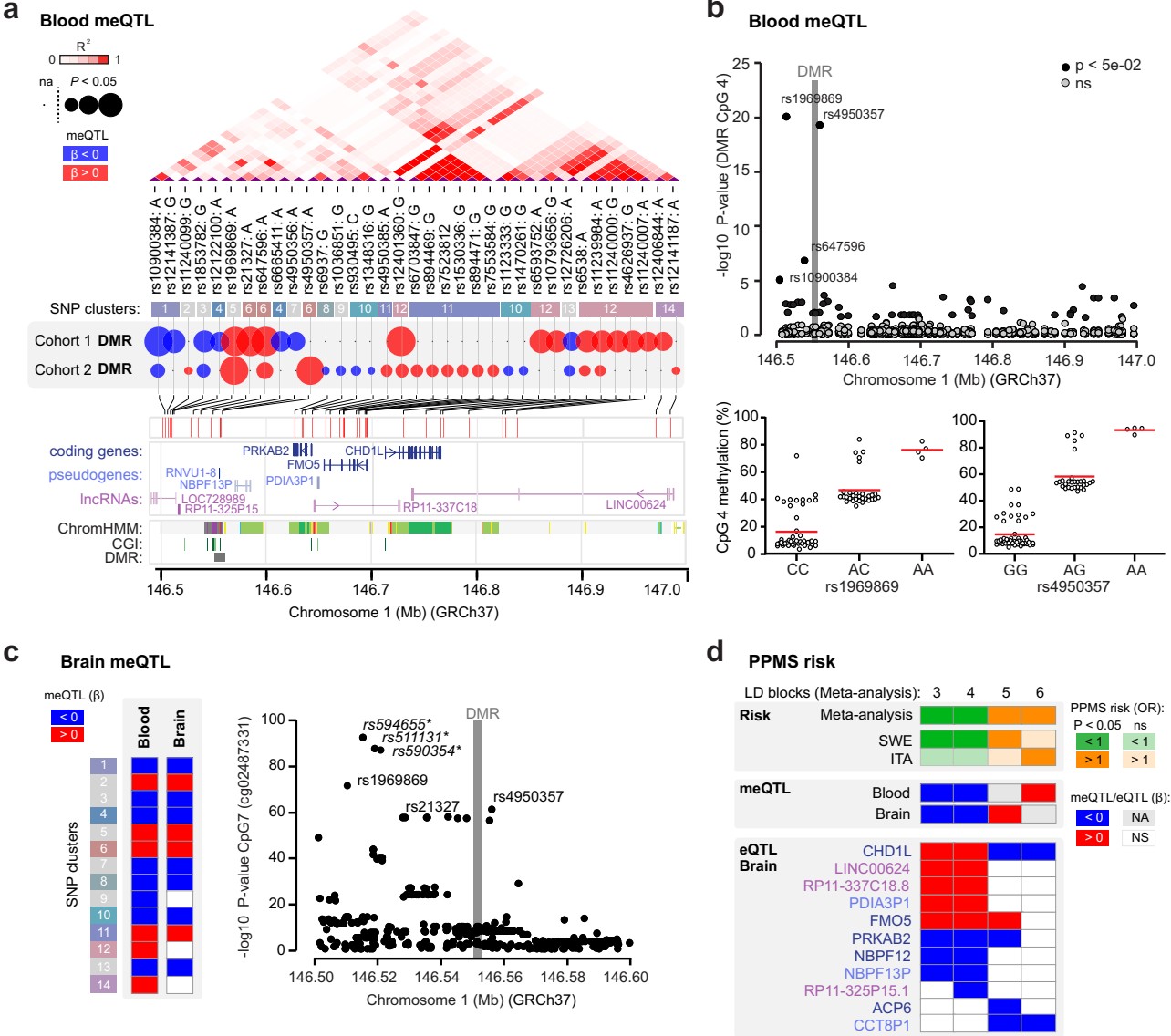

**Fig. 3 | Genetic control of methylation and gene expression in the 1q21.1 locus.**
**a** Genomic and functional annotation of the methylation-controlling SNPs. Linkage
disequilibrium (LD) is indicated by shades of red and clustering of SNPs is based on
high LD between SNPs ($R^2 > 0.8$). Colored circles show SNP effects on DMR
methylation: size positively correlates with significance (−log10 $P$-value) and blue
and red colors represent negative and positive effects of the minor allele, respec-
tively. Gene location and regulatory features, i.e. CpG island (CGI) and chromatin
state segmentation (ChromHMM), from ENCODE/Broad. **b** Association between
genetic variation at the extended locus (upper panel) and DNA methylation in
blood at CpG 4 of the 1q21.1 DMR, obtained using meQTL analysis in cohort 2
($n = 82$). Significance is represented as −log10 ($P$-value). Association of the two
strongest variants rs1969869 and rs4950357 with CpG 4 (as an example, lower
panel). CpG codes are presented in Fig. 2c. **c** Effects of SNPs within each SNP cluster
on DNA methylation at 1q21.1 DMR in the brain (using xQTL serve platform[15]) with
only significant associations displayed in colors, blue and red colors representing

negative and positive effect of the minor allele, respectively (left panel) and asso-
ciation between genetic variation in the extended locus and DNA methylation at
cg02487331 (CpG 7) (right panel). Significance is represented as −log10 ($P$-value). *
Indicates strong LD with rs21327 ($R^2 > 0.7$). **d**. Association between genetic varia-
tion, DNA methylation (using xQTL serve platform[15]), and gene expression (using
GTEx database) in the extended locus for SNPs tagging four variants (LD blocks)
displaying significant association with PPMS in the meta-analysis of Swedish (SWE)
and Italian (ITA) cohorts. Green and orange colors reflect protective and risk effects
on PPMS risk conferred by the genetic variants, respectively. Blue and red colors
represent the negative and positive effects of the minor allele, respectively, on
methylation or expression. NA not available, NS not significant. The data shown in
this panel are available in Supplementary Data 7 (meQTL, eQTL) and 8 (genetic
analysis). A linear regression model with a Bonferroni-corrected or FDR-adjusted
threshold of 0.05 was used for all the statistical tests.

databases, we found that a large fraction of the variants (mapping to
10/16 and 34/57 LD blocks from the meta-analysis and Swedish cohort,
respectively) affected transcript levels of neighboring genes within the
locus (*PRKAB2*, *CHD1L*, and *FMO5*) as well as long non-coding RNAs in the
brain (Supplementary Data 7). As exemplified in Fig. 3d, variants pre-
disposing for higher DMR methylation and PPMS risk (LD blocks 5 and 6)
associated with lower *CHD1L* (normalized enrichment score NES < −0.5)
and *PRKAB2* expression (NES < −0.3) while variants conferring low DMR

methylation and PPMS protection (LD blocks 3 and 4) associated with
higher *CHD1L* and *FMO5* expression (NES > 0.5) and lower *PRKAB2*
(NES < −0.3) transcript levels (Supplementary Data 7). The *CHD1L* gene
displayed the most consistent changes regarding the impact of genetic
variation on PPMS risk, methylation, and *CHD1L* expression. These data,
connecting genetic variation, methylation, and transcription in the
extended 1q21.1 region, are consistent with genomic annotations of the
DMR underscoring putative regulatory features as well as short-range

**Table 1 | A meta-analysis of genetic association studies with PPMS (*n* = 1104) in comparison to BOMS (*n* = 11,335)**

| SNP[a] | BP | Meta-analysis | | | | | SWE (*n* = 9850) | | ITA (*n* = 2589) | |
|---|---|---|---|---|---|---|---|---|---|---|
| | | CHR | LD# | A1 | *P* | OR | *P* | OR | *P* | OR |
| rs900347 | 145726727 | 1 | 1 | A | 0.0285 | 0.90 | 0.0352 | 0.87 | 0.3624 | 0.94 |
| rs11800992 | 146501783 | 1 | 2 | A | 0.0093 | 0.82 | 0.0260 | 0.78 | 0.1399 | 0.85 |
| rs72691008 | 146708291 | 1 | 3 | A | 0.0017 | 0.69 | 0.0083 | 0.69 | 0.0895 | 0.70 |
| rs7514808 | 146570076 | 1 | 4 | A | 0.0286 | 0.89 | 0.0367 | 0.86 | 0.3632 | 0.93 |
| rs72706463 | 146615555 | 1 | 5 | A | 0.0458 | 1.29 | 0.1141 | 1.37 | 0.2002 | 1.24 |
| rs10453880 | 146693689 | 1 | 6 | T | 0.0065 | 1.26 | 0.1083 | 1.21 | 0.0236 | 1.32 |
| rs145465008 | 146630173 | 1 | 7 | C | 0.0293 | 1.51 | 0.4867 | 1.17 | 0.0048 | 2.54 |
| **rs115153075** | 146978424 | 1 | 8 | T | 0.0006 | 1.48 | 0.0058 | 1.42 | 0.0303 | 1.74 |
| **rs12096043** | 147060953 | 1 | 9 | T | 0.0001 | 1.84 | 0.0005 | 2.03 | 0.0633 | 1.58 |
| rs672619 | 147058575 | 1 | 10 | T | 0.0013 | 1.34 | 0.0374 | 1.32 | 0.0146 | 1.35 |
| rs10494248 | 147108260 | 1 | 11 | A | 0.0079 | 1.72 | 0.0270 | 1.68 | 0.1386 | 1.82 |
| rs140566115 | 147126437 | 1 | 12 | A | 0.0244 | 1.53 | 0.0335 | 1.66 | 0.3586 | 1.34 |
| rs116430961 | 147312516 | 1 | 13 | C | 0.0101 | 1.62 | 0.1335 | 1.44 | 0.0265 | 1.93 |
| rs61740912 | 147313970 | 1 | 14 | G | 0.0012 | 1.86 | 0.0098 | 1.81 | 0.0483 | 1.97 |
| rs112044341 | 147317235 | 1 | 15 | A | 0.0193 | 1.47 | 0.0577 | 1.46 | 0.1707 | 1.49 |
| rs74123747 | 147382534 | 1 | 16 | T | 0.0496 | 1.27 | 0.2194 | 1.20 | 0.0813 | 1.50 |

*PPMS* primary progressive MS, *BOMS* bout-onset MS (RRMS + SPMS), *OR* odds ratio, *LD#* linkage disequilibrium block, *SWE* Swedish cohort, *ITA* Italian cohort, *CHR* chromosome; *P* P-value, *A1* effect allele, *BP* base pair (GRCh37/hg19).
[a]SNP ID highlighted in bold remains significant after Bonferroni correction for multiple testing (p < 7.2 ×10⁻⁴).

physical interaction with proximal chromatin segments harboring eQTL genes (as annotated by ENCODE Roadmap, IHEC, Supplementary Fig. 1). The analyses of DNA methylation levels and transcription factor (TF) binding motifs in 1q21.1 suggest that the region has a potential to exert regulatory functions in all CNS cell types, particularly neurons (Supplementary Fig. 2). Thus, the variants influencing the risk of developing PPMS and controlling DNA methylation in the 1q21.1 locus likely also exert an effect on the expression of proximal genes in the brain, particularly *CHD1L*, *FMO5*, and *PRKAB2*.

## 1q21.1 methylation regulates *CHD1L* and *PRKAB2* expression

In order to functionally address the causal relationship between epigenetic and transcriptional changes, we first tested whether methylation at the 1q21.1 DMR can exhibit regulatory properties on transcription, as suggested by genomic annotations of the locus (Fig. 2a, Supplementary Fig. 1), by examining methylation-sensitive enhancer and promoter activity using in vitro methylation assays. Two distinct 1 kb fragments harboring the DMR sequence, isolated from PPMS patients with low (rs1969869: CC) and high (rs1969869: AA) methylation levels at the identified DMR, were inserted in CpG-free vector-based reporter system. The sequences were subsequently methylated using two different methyltransferases, *M.SssI* enzyme methylating all CpG sites (57 CpGs) and *HhaI* enzyme targeting only internal cytosine residues from the consensus sequence GCGC (7 CpGs). The 3'-5' oriented DNA sequence of both genotypes exhibited potent constitutive promoter activity on the reporter gene (Mock condition), this basal activity being nonetheless halved in carriers of the minor allele compared to AA-homozygotes (Fig. 4a). The DMR manifested enhancer activity as well, although to a lesser extent, with no clear contribution of each allele or sequence orientation (Fig. 4b). Comparison of the unmethylated and methylated sequences revealed a significant reduction of the promoter and enhancer activity when the insert was fully methylated (Fig. 4a, b). Thus, controlling the level of methylation in the DMR region might have a regulatory effect on gene expression.

To formally address the impact of DNA methylation on endogenous gene expression, we exploited the CRISPR/dCas9-based epigenome editing platform which allows editing of methylation in a sequence-specific manner in combination with tailored single guide RNA (gRNA). We took advantage of the fact that several cell lines display hypermethylation at the 1q21.1 locus (as annotated by ENCODE Roadmap, IHEC), mimicking methylation pattern in PPMS patients, and developed constructs expressing deactivated Cas9 fused to the catalytic domain of the demethylating enzyme TET1 (dCas9-TET1) to remove methylation from specific CpGs (see "Methods" section for further description). We could validate the editing efficiency of dCas9-TET1 constructs by targeting 5 CpGs composing *MGAT3* gene promoter, with 10-50% demethylation as previously reported[22] (Supplementary Fig. 3). To remove methylation at CpGs of the PPMS-associated DMR, we first designed and functionally screened nine gRNAs targeting the two 450K probes exhibiting the largest changes (cg21263664, cg03526459) and adjacent CpGs in HEK293T cells (Fig. 4c). Negative controls consisted of cells transfected with a non-targeting (nt) gRNA, the catalytically inactive form of TET1 (TET1-IM) or non-transfected cells. DNA methylation analysis, using pyrosequencing, showed different degrees of demethylation ranging from 10% to 50% reduction in methylation at single or multiple CpG sites depending on the gRNA (Fig. 4d). Delivery of dCas9-TET1 together with gRNA2 and gRNA3 could achieve robust demethylation at four sites, including cg03526459 (Fig. 4d). We next adapted this strategy to neuroblastoma SH-SY5Y cells using transduction with dCas9-TET1 + gRNA2 + gRNA3 lentiviruses (Fig. 4e) and could confirm the editing efficiency in these cells with 15-35% decrease of DNA methylation spanning throughout CpGs 1–3 (Fig. 4f). SH-SY5Y cells display hypermethylation at the 1q21.1 locus (ENCODE Roadmap) mimicking methylation pattern in PPMS patients. We then assessed by qPCR the expression of genes located in the vicinity of the DMR. These include the three coding genes found regulated by the same meQTL-SNPs in the normal brain (Fig. 3d), namely *CHD1L* gene encoding a DNA helicase protein involved in DNA repair, *PRKAB2* gene encoding the non-catalytic subunit of the AMP-activated protein kinase (AMPK), which serves as an energy sensor protein kinase regulating cellular energy metabolism and *FMO5* gene encoding a Baeyer-Villiger monooxygenase implicated in liver function and metabolic ageing[23]. The induced demethylation, even moderate, of CpGs in the 1q21.1 DMR in SH-SY5Y cells resulted in increased expression of *CHD1L* and *PRKAB2* genes while the expression of *FMO5* and other proximal transcripts, *PDIA3P1* and *PFN1P5*, did not vary significantly (Fig. 4g).

Collectively, these data suggest that the intergenic PPMS-associated DNA methylation at 1q21.1 region exerts regulatory

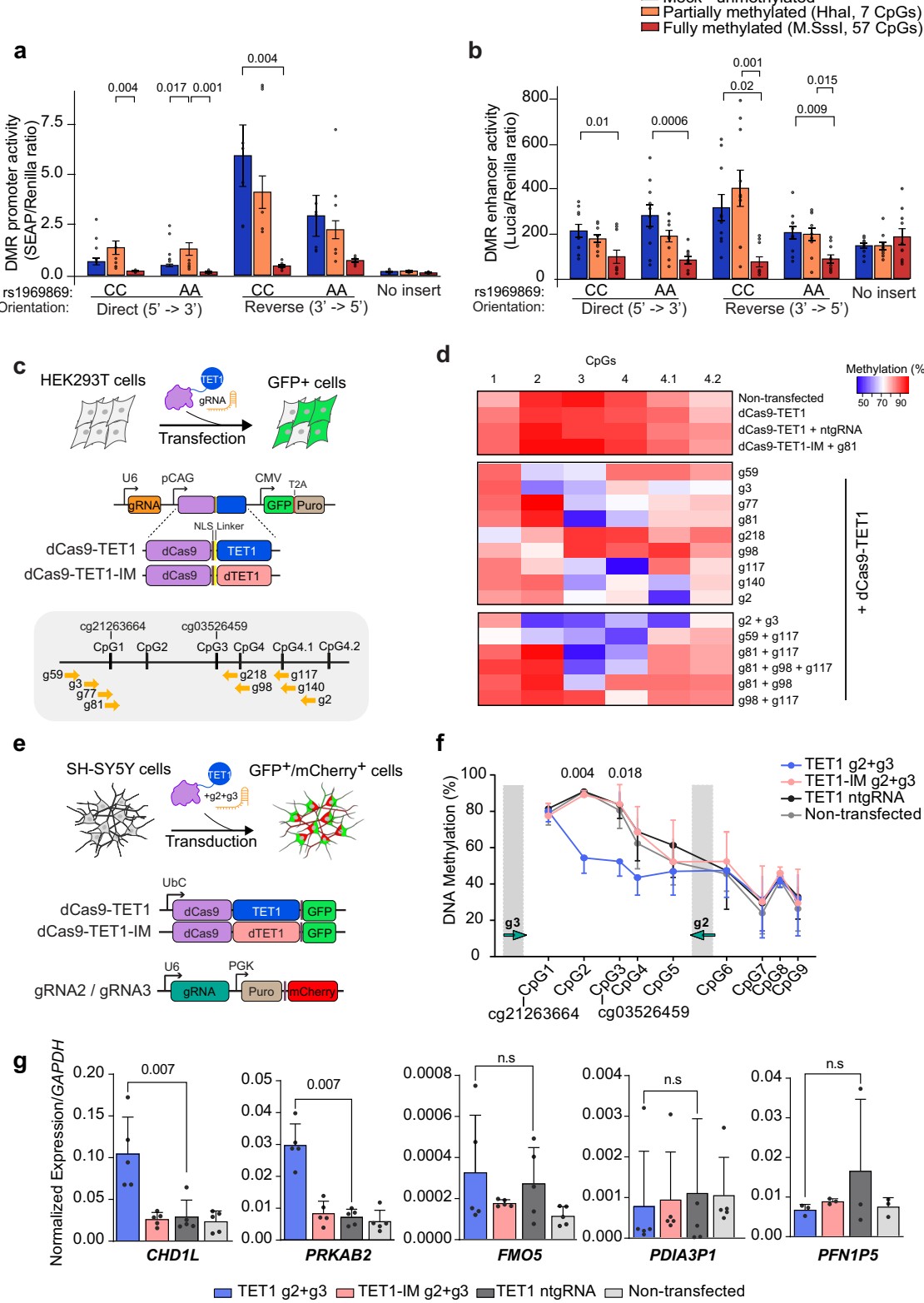

properties on transcriptional processes. The methylation-mediated regulation of gene expression affects *CHD1L* and *PRKAB2* genes in neuron-like cells.

### *CHD1L*-gene network is enriched in PPMS brain tissue

To further delineate the putative relevance of the genes included in the 1q21.1 locus in PPMS brain pathology, we constructed an unbiased correlation network analysis using RNA-sequencing-derived gene expression matrix from brain tissue samples of progressive MS patients ($n = 12$) and non-neurological controls ($n = 10$)[9] (Fig. 5). After gene-pair permutation and planar filtering, the network consisted of 0.5 million interactions among 27,059 genes (FDR < 0.05) (Fig. 5a). Multiscale clustering analysis further clustered these interactions into 757 non-overlapping modules (Fig. 5b). We then evaluated the relevance of these modules for PPMS ($n = 5$) and SPMS ($n = 7$) phenotypic traits by applying cluster trait association analysis

**Fig. 4 | Functional impact of methylation at the 1q21.1 DMR on gene expression.** **a, b** Promoter (**a**) and enhancer (**b**) activity of the DMR, using CpG-free promoter-free (SEAP) and promoter-containing (Lucia) reporter gene vectors, respectively. Constructs in direct or inverted orientation of DNA segments derived from individuals varying according to the genotype at rs1969869 were partially or fully methylated using HhaI and M.SssI enzymatic treatment, respectively. Results show relative activity ±SEM of SEAP (two experimental replicates, $n = 6$) or Lucia (three experimental replicates, $n = 10$) normalized against Renilla (2-way ANOVA with Bonferroni correction for multiple comparisons). **c** Schematic representation of the experimental design for gRNAs screen in HEK293T cells including the features of the constructs and gRNA locations. **d** Heatmap of the DNA methylation levels in successfully transfected (GFP positive) HEK293T cells three days following co-transfection of dCas9-TET1 with single or combined gRNAs in comparison to control conditions, deactivated TET1 (TET1-IM), non-targeting gRNA (ntgRNA) and non-transfected cells. **e** Schematic representation of the experimental design for functional investigation in SH-SY5Y cells. **f** DNA methylation levels in SH-SY5Y cells following delivery of dCas9-TET1-, gRNA2- and gRNA3-containing lentiviruses. Methylation percentages represent the mean ± SD of three experiments (2-way ANOVA followed by Turkey's multiple comparisons test). **g** Experiment showing expression of *CHD1L*, *PRKAB2*, *PDIA3P1*, and *FMO5* genes relative to *GAPDH* transcript levels, quantified using RT-qPCR. The expression levels represent the average of at least three experiments (mean ± SD, two-tailed Student's t-test). n.s, non-significant. Source data are provided as a Source Data file.

(Fig. 5b). Three modules survived statistical testing, one of them (module #1) being correlated to SPMS while the two others (modules #2 and #3) showing high significance to PPMS phenotype (Fig. 5b, Supplementary Data 9). Strikingly, these modules were found centered on genes from the extended 1q21.1 locus. Indeed, exploration of the biggest module (module #2), consisting of 380 nodes and 1079 interactions, revealed *CHD1L* as a central node within this network, closely interacting with six genes of a sub-module (Fig. 5c). Relative centrality could also be observed for the other two modules, which include *FMO5* among 40 nodes (module #1) and *PRKAB2* among 54 nodes (module #3) (Fig. 5c). Of note, *CHD1L* and *PRKAB2* were the only genes from the extended 1q21.1 locus to partake in the regulatory gene network underlying PPMS brain pathology in our analysis. Gene ontology analysis, using Ingenuity Pathway Analysis, underscored the implication of all modules in nervous processes. This is particularly the case for module #2 with terms linked to neuronal functions, while additional *Biological Functions* were found associated with module #1 (e.g. metabolic processes) and module #3 (e.g. cell cycle) (Fig. 5c, Supplementary Data 10). Thus, findings from an unbiased approach support a potential role of genotype-methylation-expression regulatory network affecting genes of chromosome 1q21.1 extended locus, such as *CHD1L* and *PRKAB2*, in CNS processes in PPMS patients.

To verify the network modules, we used a large GTEx dataset from healthy individuals (Supplementary Data 11). We did not detect our networks in any of the 14 strata, but *CHD1L*, *FMO5*, and *PRKAB2* genes belonged to significantly co-expressed gene networks only in the cortex and tibial nerve. This could indicate that, beyond constitutive expression of these genes in multiple tissues and cells, the existence of the networks is dependent on the disease processes and possibly more restricted to neurons. We thus investigated additional datasets comprising CNS tissue from MS patients, particularly focusing on those in which co-expressed networks containing *CHD1L*, *FMO5*, or *PRKAB2* gene were detectable (Supplementary Data 11). The CUX2+ excitatory neurons, previously shown to be specifically affected in MS, displayed significant co-expression of *CHD1L*, *FMO5*, and *PRKAB2* with other genes, thereby forming network modules, in the single nuclei RNA-sequencing (snRNA-seq) dataset from the original study[24]. We detected a significant overlap of our *CHD1L* module, but not *FMO5* and *PRKAB2* modules, with the CUX2+ specific gene signature (overlapping = 18, *p*-value = $2.95 × 10^{-6}$, OR = 3.65). Conversely, the *CHD1L* module in CUX2+ neurons significantly overlapped with our *CHD1L* module with shared closest interacting genes and pathways related to neuronal functions (overlapping = 20, *p*-value = 0.03, OR = 1.56, Supplementary Data 9, Supplementary Fig. 4).

Thus, we detected a significant overlap between PPMS-associated *CHD1L*-gene network and the gene signature of neurons previously reported to be specifically affected in MS[24], which, together with the concordant meQTL and eQTL effects of PPMS-associated variants for this gene, implicate *CHD1L* gene in neuronal vulnerability in PPMS.

## Dysregulation of *CHD1L* affects neuronal development and activity

Previous studies unveiled the essential role of *CHD1L* at early embryonic developmental stages, with its absence at the zygote stage causing early developmental arrest in rodents[25,26]. To investigate the role of *CHD1L* in cellular functionality relevant for MS pathology in vivo, we utilized a *chd1l* mutant zebrafish line and examined whether dosage-dependent *chd1l* expression affected cells of the neuroglial lineage in the head and the caudal part of the zebrafish larvae (Fig. 6a). Four days post-fertilization (dpf), *chd1l*+/− mutant larvae displayed a significant reduction in the number of axonal tracts projecting from the optic tecta compared to control *chd1l*+/+ larvae (Fig. 6b, mean axonal tracts = 12.2 vs. 10.7 respectively, $p = 8.7 × 10^{-6}$). Of note, mutant larvae did not exhibit significant alteration of sprouting and/or path-finding of peripheral neurons (Supplementary Fig. 5a, d). We next used the *Tg(olig2:EGFP)* reporter line that marks oligodendrocyte precursors and motor neurons at 3 dpf[27,28] and found a significant decrease of the number of olig2-expressing cells in the hindbrain (Fig. 6c, normalized mean = 0.19 vs. 0.16 in *chd1l*+/+ and *chd1l*+/− respectively, $p = 4.1 × 10^{-6}$) and dorsally migrating from the spinal cord (Fig. 6d, normalized mean = 36.31 vs. 29.42 in *chd1l*+/+ and *chd1l*+/− respectively, $p = 8.1 × 10^{-3}$) of *chd1l* mutant larvae. These findings indicate that *chd1l* contributes to CNS development and integrity in vivo.

To further examine the role of *CHD1L* in neuronal function specifically, we used iPSC-derived neural progenitor cells (NPCs) patterned via dual SMAD inhibition and differentiated them into forebrain neurons for five weeks (DIV 35) in the presence of *CHD1L* siRNAs (knock-down, KD) or non-targeting (NT) siRNAs (Fig. 6e). The NPCs expressed markers of early neuroectodermal induction (PAX6, SOX1) and following differentiation, neurons exhibited MAP2+ neurites marked by putative synapses (SYN1) (Fig. 6f). The successful knock-down of *CHD1L* during differentiation did not affect the expression of intermediate progenitor and migration markers (*EOMES*, *NCAM1*), cortical layer markers (*CTIP2*, *SATB2*) or immature neuronal marker (DCX) (Fig. 6g). Instead, *CHD1L*-deficient neurons presented branching abnormalities reflected by smaller MAP2-positive neuritic protrusions, while the synaptic density seemed unaffected (Fig. 6h). *CHD1L*-knock-down also resulted in neuronal functional impairment detectable as reduced calcium intensity signal, overall suggesting weaker electrical activity in neurons expressing lower levels of *CHD1L* (Fig. 6i, j).

Altogether, our data imply that dysregulation of *CHD1L* in CNS may lead to functional neuronal impairment.

## Discussion

MS is a multifactorial and heterogeneous neurological disease with unpredictable trajectories affecting young individuals. Due to our partial knowledge of the pathological processes underlying disease progression, treatment of progressive MS remains the greatest challenge in patient care. In this study, we utilized a multi-omics (genetic, epigenetic, transcriptomic) approach in several cohorts in

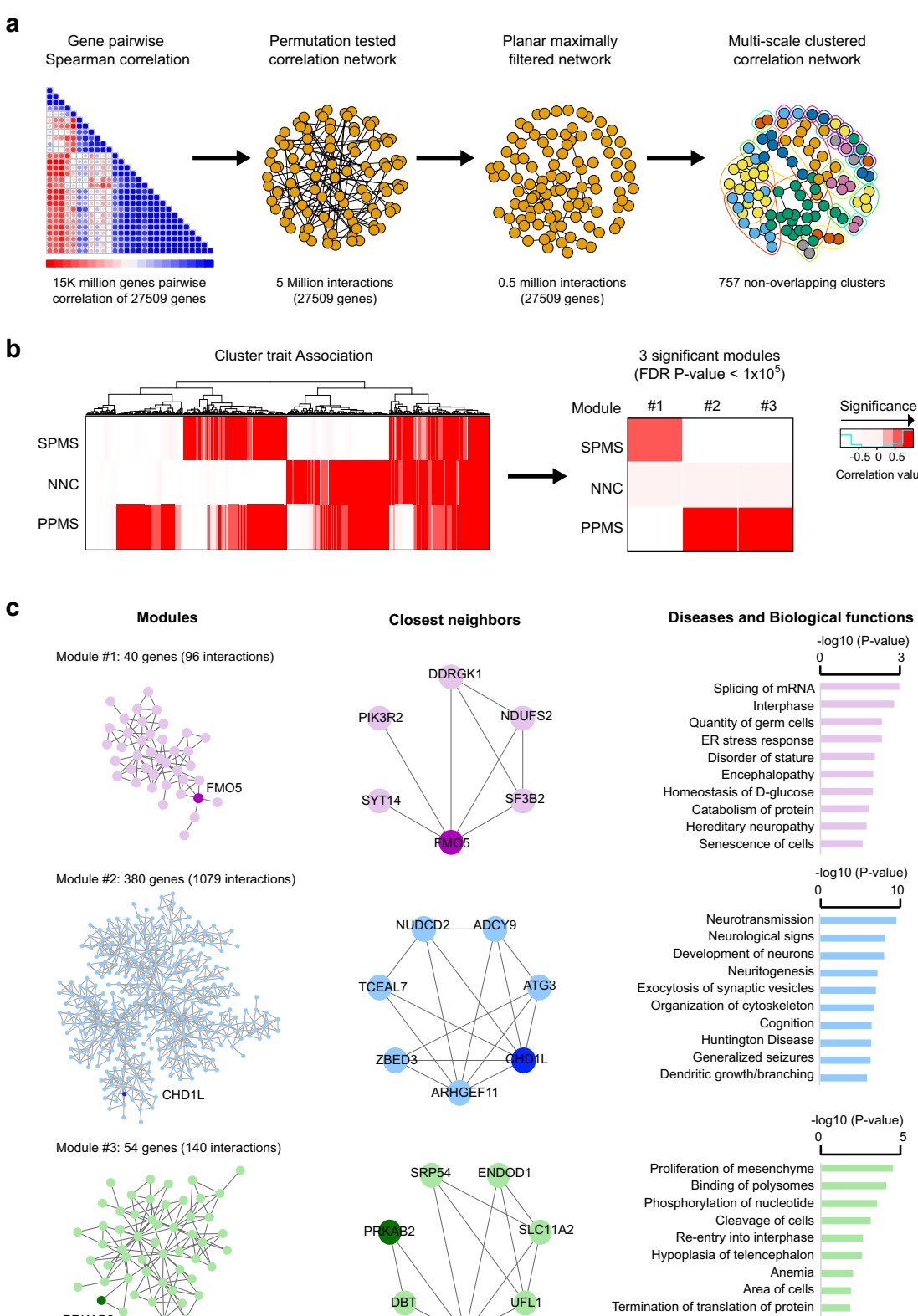

combination with in-silico, in vitro, and in vivo functional experiments and analysis to gain more insights into the primary progressive form of MS. We identified a genetically controlled hypermethylated region in the chromosome 1q21.1 locus which might affect the expression of proximal genes, particularly *CHD1L*, in the brain and neurons of PPMS patients. Dysregulated *CHD1L* expression, in turn, altered neuronal development and oligodendrogenesis in zebrafish

and caused structural and functional abnormalities in human neurons.

Attempts to identify genetic variants that predispose for PPMS have not yielded reproducible associations with the exception of the known association with *HLA-DRB1*15:01*[29,30]. One of the challenges of the genetic studies in PPMS is the limited availability of PPMS cases that comprise around 10% of all individuals affected with MS. We and

**Fig. 5 | Implication of *CHD1L* and *PRKAB2* genes in PPMS brain pathology.**
**a** Schematic representation of the correlation network analysis using bulk MS brain transcriptome[9]. The different steps are depicted from left to right: correlation matrix containing gene-gene pairwise Spearman correlations; illustration of the permutation test filtered correlation network surviving an FDR *P*-value < 0.05 and the planar maximally filtering to convert the scale-free network to be able to overlay on a plane spherical surface, reducing the network to 0.5 million interactions among 27,509 genes and, finally; representation of the multiscale clustering of the network resulting into 757 non-overlapping clusters. **b** Heatmap of the correlation coefficients of all 757 non-overlapping modules (left) and the three modules surviving correlation FDR *P*-value < 0.05 (right) to each tested phenotypic trait ($n_{PPMS} = 5$, $n_{SPMS} = 7$ and $n_{non-neurological\ controls} = 10$), with red gradient colors representing negative (not significant) to positive (significant) correlation. Spearman correlation is applied on every gene pair and *P*-values were adjusted for multiple comparisons by FDR (< 0.05). **c** Representation of the three modules (left), the closest neighbors to candidate genes in each module (middle), and Gene Ontology findings, i.e. *Disease and Biological functions*, obtained using Ingenuity Pathway Analysis (Fisher's exact test, *P*-value < 0.05). MS multiple sclerosis, PPMS primary progressive MS, SPMS secondary progressive MS, NNC non-neurological controls.

others have shown that integrating additional layers of gene regulation, such as DNA methylation, can reveal genetic associations that are difficult to identify using conventional genetic studies[11,31–33]. Here, we combined methylome profiling with locus-specific genetic association analysis in independent cohorts and identified a genetically controlled DMR in the 1q21.1 locus that displayed hypermethylation in PPMS patients specifically.

The same genetic variants that control methylation and gene expression in the locus are associated with PPMS risk, suggesting a functional link between genetic, epigenetic, and transcriptional modalities in 1q21.1 in PPMS patients. We provided evidence for association with the risk of developing PPMS in two independent cohorts from Sweden and Italy comprising 1104 PPMS cases compared to 11,335 relapse onset MS, which is the largest reported PPMS cohort to date. Nevertheless, establishing unequivocal genetic association alone would require cohort sizes that could be challenging to obtain even by large multicentric efforts, further emphasizing the need to combine multiple independent layers of evidence.

Our findings should also be considered in light of the remarkably complex genomic architecture of the 1q21.1 locus displaying considerable copy-number variations (CNVs). Indeed, the extended 1q21.1 genomic locus covering the identified DMR is characterized by the largest copy-number expansion between non-human and human primate lineage, such variation correlating with evolutionary brain size in a dose-dependent manner[16–18]. CNVs in this region have also been associated with brain disorders such as cognitive and motor deficits, neuroblastoma, autistic spectrum disorder, and schizophrenia[19–21,34,35]. Interestingly, findings from in vitro neuronal models, i.e., iPSC-derived neurons[34] and hES-derived cortical organoids[36], and in vivo rodent models[37] jointly show that deletion and duplication of the 1q21.1 region in neurons substantially impact various neurodevelopmental characteristics, such as proliferation, differentiation potential, neuronal maturation and synaptic density. Our further analysis revealed that some of the SNPs conferring risk for PPMS, are associated with Schizophrenia[38] and are controlling the expression of neighboring genes including *CHD1L* in the brain cells. These associations with brain size and neurodevelopmental disorders further support the idea that inherent brain tissue vulnerability influenced by the genes in the locus may predispose for rapid and progressive MS disease, as recently suggested for the severity of MS[39].

The dual meQTL and eQTL effects observed in the CNS tissue, underscore the possible contribution of genetic-epigenetic-transcriptional regulation in PPMS brain pathology. In this context, we found that genetic variation associated with elevated DNA methylation levels at 1q21.1 and increased PPMS risk is linked to lower expression of *CHD1L* transcript in the CNS. Inversely, PPMS protective variants are associated with low 1q21.1 methylation and high *CHD1L* expression. Such a coherent reciprocal effect could not be observed for *PRKAB2,* likely due to cellular heterogeneity of bulk brain tissue masking possible cell type-specific effects. Among the potential mechanisms suggested to mediate genetic control of methylation, SNPs can influence DNA methylation by directly disrupting CpG sites, altering transcription factor binding sites, and/or influencing the three-dimensional chromatin structure[40–42]. Our annotation of the 1q21.1 DMR showed that the CpG sites were not affected by local SNPs and that chromatin looping exists between the DMR and the region harboring the risk SNPs. Yet, whether such potential physical interaction directly influences the DMR accessibility to epigenetic modifiers such as methylating enzymes or whether the genetic control engages other mechanisms remains to be elucidated.

Undeniably, given the occurrence of meQTL effects in the blood, one cannot exclude a contribution of the 1q21.1 regulatory network in the peripheral immune compartment as well. Similar trans-tissue effects with shared patterns between the immune and nervous compartments have been previously observed in the context of progressive MS disease[43]. Collectively, these findings portray DNA methylation as a putative intermediary of genetic influence on gene expression, particularly in the CNS of PPMS patients.

We next sought to formally explore the possibility that DNA methylation may directly influence gene expression. We first demonstrated the intrinsic methylation-sensitive promoter and enhancer activity of the PPMS-associated DMR sequence using in vitro methylation assays. We then utilized the CRISPR/dCas9-TET1 epigenome editing approach to achieve effective and locus-specific demethylation at CpGs of the identified DMR and found that manipulating DNA methylation at the 1q21.1 DMR affects the expression of the proximal *CHD1L* and *PRKAB2* genes in neuron-like cells. These data, together with the indisputable involvement of the 1q21.1 region in neuronal circuitry and brain integrity[16,17,19–21], implicate genes of the locus in neuropathological processes in PPMS.

Our unbiased correlation network analysis of published transcriptomic data obtained from post-mortem brain tissue from PPMS, SPMS patients, and controls[9], identified *CHD1L* and *PRKAB2* transcripts as implicated in PPMS-specific regulatory gene networks. *CHD1L* encodes a DNA helicase implicated in chromatin-remodeling during DNA repair, with both depletion and overexpression of *CHD1L* leading to genomic sensitivity to DNA-damaging agents[44]. *PRKAB2* encodes the beta subunit of AMP-activated kinase, a neuroprotective energy-sensing kinase exerting pivotal roles in the cellular energy metabolism required for the maintenance of neuronal integrity[45]. Interestingly, bulk and single-cell transcriptomic analysis in brain tissue have found dysregulation of several genes located within the identified 1q21.1 locus, including *CHD1L* and *PRKAB2* transcripts along with several members of the *NBPF* family of genes, in progressive MS patients compared to control individuals[9,24]. Markedly, the *CHD1L* gene has been identified in the stress-related signature of excitatory CUX2-expressing projection neurons in upper-cortical layers, which are particularly susceptible to degeneration in MS patients[24]. Additionally, a large fraction of *CHD1L* (3/6) closest network neighbors identified in our study were found significantly upregulated in these stressed excitatory neurons as well. Moreover, our *CHD1L* network module displayed a significant overlap, both on the gene and pathway level, with the transcriptional signature of CUX2+ neurons as well as the *CHD1L* network detected in CUX2+ neurons using snRNA-seq dataset from Schirmer et al.[24]. It is therefore reasonable to speculate on the involvement of *CHD1L* in the compensatory mechanisms

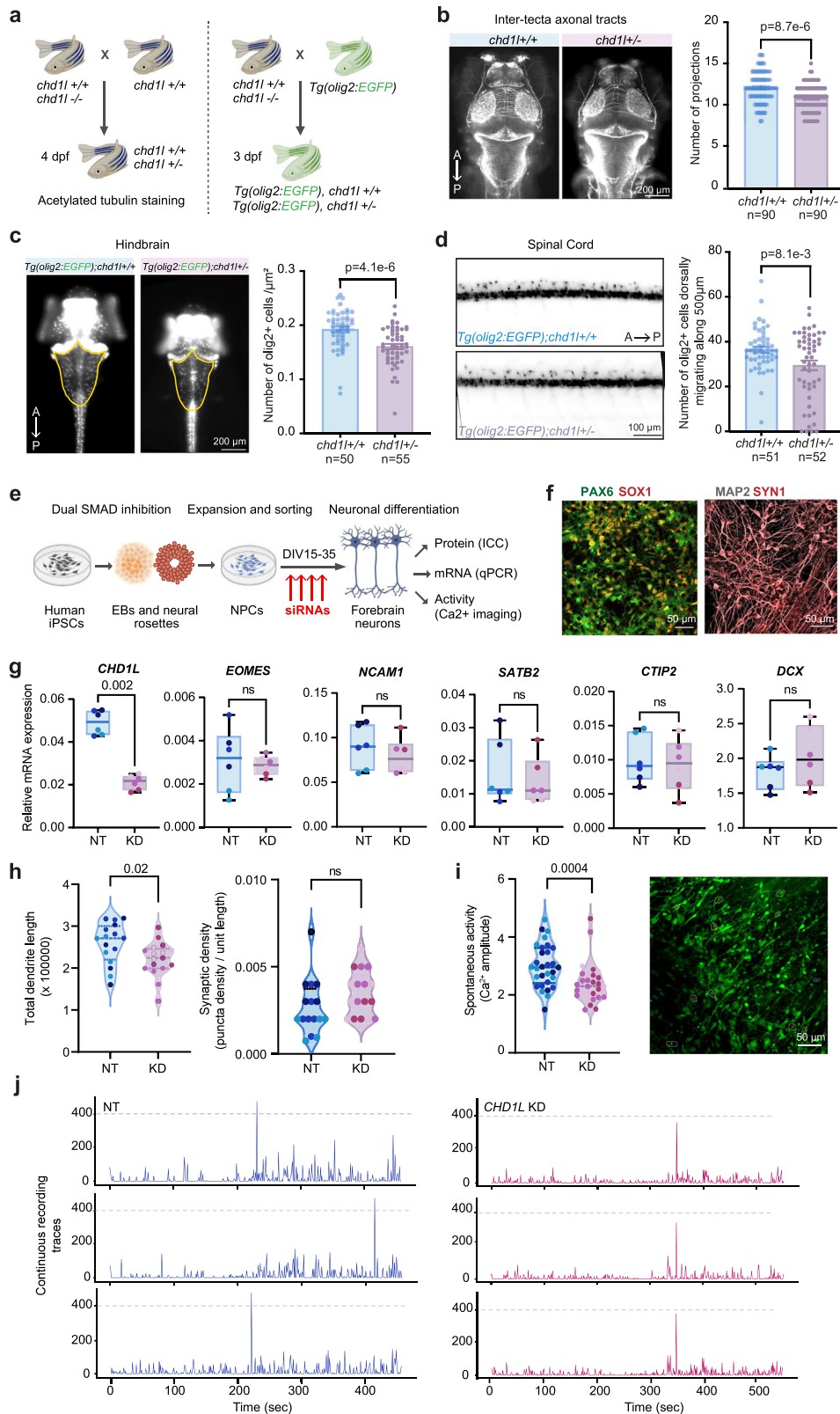

counteracting the cellular burden, e.g., genomic and metabolic stress, observed in these neurons, although validation in large PPMS dataset is warranted. More specifically, genetic predisposition to high methylation and low expression of the *CHD1L* gene might influence susceptibility to PPMS by enhancing neuronal vulnerability and limiting the inherent repair capacity in the brain.

Findings from our study converge to several lines of evidence, namely the concordant meQTL/eQTL effect in the CNS, methylation-mediated *CHD1L* regulation, and detection of dysregulated *CHD1L*-gene network pertaining to neuronal processes in the PPMS brain, implicating *CHD1L* in neuronal vulnerability in PPMS. To formally address the role of *CHD1L* in neuronal function, we conducted func-

**Fig. 6 | Functional impact of low *CHD1L* expression in neurons and oligodendrocytes. a** Schematic of zebrafish experimental design. **b** Dorsal view of control and *chd1l*+/− larvae at 4 dpf stained with acetylated tubulin. Barplot of the inter-tecta axonal tract projections count of 4 dpf control and *chd1l*+/− larvae (Wilcoxon test, mean ± SEM, *n* = 3 replicate/genotype, 30 larvae/replicate). **c** Dorsal view of *Tg(olig2:EGFP);chd1l*+/+ and *Tg(olig2:EGFP);chd1l*+/− at 3 dpf with the hindbrain used for olig2-positive cells count outlined in yellow. Barplot of the number of olig2-positive cells per μm2 in the hindbrain of *Tg(olig2:EGFP);chd1l*+/+ and *Tg(olig2:EGFP);chd1l*+/− larvae at 3 dpf (Wilcoxon test, mean ± SEM, *n* = 3 replicate/genotype, 12–20 larvae/replicate). **d** Lateral view of the spinal cord of *Tg(olig2:EGFP);chd1l*+/+ and *Tg(olig2:EGFP);chd1l*+/− larvae at 3 dpf. Barplot of the number of olig2-positive cells migrating dorsally from the spinal cord of *Tg(olig2:EGFP);chd1l*+/+ and *Tg(olig2:EGFP);chd1l*+/− larvae at 3 dpf (Student's T-test, mean ± SEM, *n* = 3 replicate/genotype, 15–20 larvae/replicate). **e** Schematics of iPSCs experimental design. **f** Representative confocal images of markers of neuroectodermal NPCs (PAX6, SOX1) and differentiated neurons (MAP2, SYN1), all

performed in duplicates in three independent subject lines. **g** qPCR data of *CHD1L*, *EOMES*, *SATB2*, *CTIP2* and *DCX* transcripts in non-targeting (NT) and *CHD1L* knockdown (KD) neurons. Box plots indicate the median (line), 25th and 75th percentile (box), min and max (whiskers). All data points correspond to experimental duplicates in three independent subject lines (different colors, two-sided Mann–Whitney U test). **h** Quantification of neurite length (μm) and synaptic density in NT and KD neurons (two-sided Mann–Whitney U test). **i** Quantification of background-corrected changes in calcium-sensitive dye fluorescent intensity in NT and *CHD1L* KD neurons. A representative confocal image is shown. Data points correspond to averaged active regions (ROIs, 10 cells) per field of view (two-sided Mann–Whitney U test). Five to six fields of view were included for each subject line and condition. **j** Representative calcium imaging traces from NT and *CHD1L* KD neurons (3 ROIs). A anterior, P posterior, dpf days post-fertilization, NT non-targeting siRNA, KD *CHD1L*-targeting siRNA. n.s non-significant. Source data are provided as a Source Data file.

tional experiments using *CHD1L*-targeted knock-down in vivo in the zebrafish and in vitro human iPSC-derived neurons. While specific abrogation of *chd1l* in rodents is lethal at the zygote stage and impairs neuroectodermal development at the early embryonic stage[26], the data from our study further delineate a specific effect of *chd1l* on the axonal tract, neurite projection, and neuronal activity in vitro and in vivo. Such an effect aligns with the notion that neuro-axonal damage and ensuing network dysconnectivity contribute to neurodegeneration in MS[46,47], with greater axonal loss being observed in the PPMS brain compared to SPMS[48]. Yet, *CHD1L* is likely to influence the functionality of other cell types involved in the CNS development, such as oligodendrocyte lineage as suggested by results in the zebrafish, and one cannot exclude additional influences of other genes from the 1q21.1 locus, such as *PRKAB2*[45].

In conclusion, our study suggests that PPMS patients display distinct molecular changes compared to SPMS, RRMS, and control individuals. The data further support the hypothesis of a causal genetic-epigenetic-transcriptional interplay within the extended 1q21.1 locus, with functional evidence of contribution from *CHD1L* in progressive MS pathology, possibly via a detrimental effect on axonal and dendritic projections and neuronal activity. Given the reversible nature of DNA methylation, our findings open new avenues for the development of therapeutic strategies, such as targeted epigenetic therapy, in the treatment of progressive MS.

## Methods
### Ethics approval and consent to participate
All experiments on human subjects were approved by the Regional Ethical Review Board in Stockholm and carried out in accordance with institutional guidelines. Written informed consent was obtained from all study participants. iPSC experiments are in accordance with IRB approval from the Ethical Review Boards in Stockholm, Sweden. Zebrafish experiments were carried out according to the guidelines of the Ethics Committee of IGBMC and ethical approval was obtained from the French Ministry of Higher Education and Research under the number APAFIS#15025-2018041616344504.

### Cohorts
Details of the cohorts are described in Supplementary Data 1 and Fig. 1. Briefly, for genome-wide DNA methylation and meQTL analysis used in cohort 1, peripheral blood samples were collected form 140 MS, patients including 120 RRMS, 4 PPMS, and 16 SPMS patients, and 139 healthy individuals, as previously described[11]. An independent cohort, cohort 2, consisting of 48 RRMS and 36 PPMS patients (matched for age, sex, disease duration, and Swedish descent) was used for pyrosequencing validation and locus-specific meQTL analysis[49]. Cohort 3 and the additional Italian cohort, used for genetic association study, comprises 1104 PPMS and 11,335 BOMS, as described in the

corresponding sections below. Gene expression data (RNA-sequencing) from bulk brain tissue samples of progressive MS patients (5 PPMS and 7 SPMS) and non-neurological controls (*n* = 10), previously described[9], were used for correlation network analysis and validation was performed in neuronal snRNA-seq data[24]. Additionally, we utilized publicly available databases from xQTL serve[15] (*n* = 543 bulk prefrontal cortex samples, http://mostafavilab.stat.ubc.ca/xQTLServe/) and GTEx (selecting all available nervous tissues, https://gtexportal.org/) platforms to address meQTL and eQTL effects in the CNS. No sample size calculation was performed for the cohorts involved in this study. We did not explore sex-specific MS effects, however, self-reported, often validated using molecular data, biological sex was used as a covariate in all analyses.

### Genome-wide DNA methylation and meQTL analyses (cohort 1)
**DNA methylation analysis.** The methylation data from 450K arrays was preprocessed as previously described[11] and in supporting information using the Illumina default procedure implemented in the Bioconductor minfi package[50]. To identify DMRs associated with the PPMS phenotype, we used the bumphunter function in minfi package[50] with adjustment for confounders: age, self-reported sex, self-reported smoking status (ever smoker vs. never smoker), hybridization date, and the first two principal components of estimated differential cell counts. The region that has a family wise error rate (FWER) less than 0.05 with 1000 resamples and contains at least 2 probes was identified as a trait-associated DMR.

**Methylation QTL analysis.** To identify potential genetic dependency, the PPMS-associated DMR-CpGs were tested for association with genotype (594,262 SNPs) using an additive minor-allele dosage model. Genotype-DMR associations were corrected for multiple testing using a stringent Bonferroni-adjusted threshold of 0.05.

### Locus-specific DNA methylation and meQTL analyses (cohort 2)
**DNA methylation analysis.** For validation of the identified PPMS-associated DMR, pyrosequencing analysis of 7 CpG sites in the locus was performed as detailed in supporting information and Supplementary Data 12, Supplementary Fig. 6.

**Methylation QTL analysis.** Methylation data was RANK transformed in R using the R Core team (Vienna, Austria, https://www.R-project.org/). Genotyping was carried out at deCODE (deCODE genetics/Amgen, Reykjavik, Iceland) using Illumina OmniExpress chip with 716,503 SNPs mapped to the Human Assembly Feb.2009 (GRCh37/hg19). Of 84 individuals, 83 were genotyped in deCODE and 82 of them passed QC. We performed meQTL analysis of chromosome 1 from bp 146500000 to bp 147000000, in PLINK[51] excluding SNPs with less than 98% genotyping rate and SNPs that were not in Hardy-Weinberg equilibrium

(p < 0.05) and corrected for 5 population-based (ancestral informative markers) principal component analysis covariates. After quality control, 123 SNPs remained in the region. Genotype-CpG associations were corrected for multiple testing using a stringent Bonferroni-adjusted threshold of 0.05.

## TF binding evaluation

We utilized the JASPAR core database[52] through the UCSC Genome Browser track data hub to identify putative TFs with binding sites in the 1q21.1 DMR region, using default criteria. We then manually annotated and filtered for TFs with known functions in the development of neuronal and glial cells.

## Methylation levels across brain cells

We investigated the methylation levels of the 1q21.1 DMR-CpGs in neuronal and non-neuronal cell types by extracting previously published llumina array-derived data (described in detail in Kular et al.[10,12]) generated by sorting neuronal (NeuN+, n = 34) and non-neuronal/glial (NeuN-, n = 56) nuclei from post-mortem human brain samples. Additionally, individual brain cell types were sorted and/or enriched using primary in vitro cultures (purchased from Celprogen, catalog #36058DNA, 37089DNA, 36055DNA, and 736055-22DNA). They include ex vivo sorted neuronal nuclei (n = 3), in vitro and ex vivo samples of oligodendrocytes (OL, n = 7) and microglia (n = 5), and exclusively in vitro cultured oligodendrocyte progenitor cells (OPC, n = 4) and astrocytes (n = 4), with methylation data available in GEO database under the accession number GSE166207.

## Genetic association study in the Swedish (SWE) cohort

Patients from the Swedish (SWE) cohort were genotyped in two different batches at deCODE Genetics using Illumina Human OmniExpress 24 v1 (OE) and Global Screening Array MD 24 v2 (GSA) arrays, following the manufacturer's instructions. For the genetic association analysis we extracted all the imputed SNPs (n = 3057) in the extended chr.1 locus (from bp 146500000 to bp 147000000). A total of 7,682,164 autosomal variants passed quality controls and were tested for SNP-to-phenotype association (PPMS = 603 versus BOMS = 9247) in the generalized linear model including covariates, separately for individuals genotyped on OE chip and those genotyped on GSA arrays (see details in supporting information). Subsequently, the results from the two independent association studies (OE and GSA) were meta-analyzed using fixed-effect and random-effect models as implemented in PLINK. Sixty-nine LD blocks were identified and used in a Bonferroni correction to account for multiple testing.

## Genetic association study in the Italian (ITA) cohort

For the Italian cohort, patients were recruited at the Laboratory of Human Genetics of Neurological Disorders at the San Raffaele Scientific Institute in Milan, Italy, and genotyped on Illumina platforms. After quality controls (see supporting information), a logistic regression model, as described for the SWE cohort, was used to study the association between the SNPs in the extended chr.1 locus and the course of MS in a total of 2589 patients (PP = 501; BOMS = 2088).

## SWE and ITA meta-analysis

A fixed-effect model meta-analysis of the standard errors of the odds ratio, as implemented in Plink[53], was applied to the three cohorts. The number of common variants in all the cohorts was 2676. Multiple testing issue was addressed as described for the SWE cohort (see also supporting information).

## In vitro methylation assay

To address the regulatory features of the identified DMR, we used in vitro DNA methylation reporter assay. A 927 bp fragment encompassing the identified DMR was amplified using blood genomic DNA from PPMS patients presenting with low (rs1969869: CC) and high (rs1969869: AA) methylation levels at the identified DMR. The amplified products in direct and reverse orientation were inserted into a pCpG-free promoter vector (Invivogen) containing a Lucia luciferase reporter and into a pCpG-free basic vector (Invivogen) containing a murine secreted embryonic alkaline phosphatase (mSEAP) reporter gene for assessment of enhancer and promoter activity, respectively. After complete, partial, or mock methylation of the constructs and transfection of HEK293T cells (Supplementary Fig. 7), Lucia or SEAP signals were normalized against Renilla (triplicate) and experiments were replicated at least two times. Details are provided in supporting information.

## CRISPR/dCas9-TET1 epigenome editing

**dCas9-TET1 and gRNA generation.** To address dCas9-TET1-mediated epigenome editing in the SH-SY5Y cell line, we utilized the lentivirus version of the cassettes. Detailed methods to engineer the P3-dCas9-TET1-GFP-Puro (Addgene #190728), P3-Lenti-dCas9-TET1-GFP (Addgene #190729), and P3-Lenti-dCas9-TET1-IM-GFP constructs are available in supporting information. Details and maps of the final constructs used in this study are presented in Supplementary Fig. 8. All gRNAs were designed by CRISPOR Version 4.98[54] both on the sense and antisense strands, with sequence and mapping presented in Supplementary Data 12 and Supplementary Fig. 9.

**dCas9-TET1 delivery to HEK293T and SH-SY5Y cells.** To test the efficiency of epigenome editing, we exploited ease of transfection of HEK293T (female human embryonic kidney epithelial cell line[55]) for gRNAs screen. Different gRNAs were transfected either individually or in combination based on the target site, using Lipofectamine 3000 (Invitrogen). For all experiments on HEK293T and SH-SY5Y cells, DNA was extracted after 72 h and bisulfite conversion was performed using 200 ng of the extracted DNA (BS-DNA, EZ DNA methylation kit, ZYMO research). DNA methylation was assessed using pyrosequencing, as described above. For all experiments conducted on SH-SY5Y cells (female human bone marrow epithelial cell line[56]), we delivered a mix of the two gRNAs that showed the highest efficiency in reducing methylation in HEK293T cells (gRNA #2 and #3). Lentivirus generation and subsequent transduction of SH-SY5Y cells are performed based on the standard protocols (details are available in supporting information).

## qPCR analysis

Total RNA and DNA were extracted using AllPrep DNA/RNA Kits (Qiagen) according to the manufacturer's instructions. RNA and DNA concentrations and quality were verified by QIAxpert (Qiagen). Reverse transcription of RNA was performed using iScript™ cDNA Synthesis Kit (Bio-Rad Laboratories, Inc., CA) following the manufacturer's instructions. Real-time PCR was performed on a Bio-Rad CFX384 Touch Real-Time PCR Detection System using iQ™ SYBR® Green Supermix (Bio-Rad Laboratories, Inc., CA) in a standard three-step PCR protocol. The relative expressions of the selected genes were normalized to the reference gene *GAPDH*. The specificity of real-time PCR reaction was verified by the melt curve analysis. The expression level of selected genes was analyzed using ΔΔCT method[57] and compared via independent t-test. All statistical analyses were performed in GraphPad Prism 6 and 7 (GraphPad Software).

## Correlation network analysis in MS brain

**Raw data analysis.** The fastq files corresponding to bulk gene expression (RNA-sequencing) data from brain tissue samples of progressive MS patients ($n_{PPMS}$ = 5, $n_{SPMS}$ = 7) and non-neurological

controls ($n = 10$)[9,24] were extracted from the RAW RNA sequence files and checked for quality control using multiqc software to make them ready for alignment[58]. After trimming using trimgalore[59], fastqc files were aligned and annotated using STAR aligner and Stringtie software[60] by applying human hg38 refseq information from UCSC. The analysis was performed on the extracted count matrix using bash and Python.

**Network analysis.** In order to utilize a brain-specific network module, we applied a previously established bioinformatic pipeline utilizing co-expression network analysis[61], as described in supporting information. The final network consisted of 0.5 million interactions among 27,059 expressed genes which clustered into 757 non-overlapping modules. For the validation data, we applied the same pipeline on several datasets (Supplementary Data 11). In the CUX2[+] neuronal snRNA-seq count data[24], planar maximal filtration of the Spearmen correlated network of 10,780 genes was multiscale clustered using the MEGENA package in R. This resulted in 91 modules out of which 1 module with *CHD1L* was significant and was further analyzed with Fisher enrichment test and pathway analysis using clusterProfiler.

**Cluster trait association analysis.** Principal component analysis (PCA) was first performed for each cluster. Next, the correlation between the first principal component and each trait was computed as cluster relevance to the trait. The 757 clusters identified from the correlation network were evaluated for their relevance to PPMS, SPMS, and control phenotypes. Three clusters passed the FDR *P*-value < 0.05.

**Zebrafish *chd1l* knock-out experiments**
Zebrafish (Danio rerio) were raised and maintained as described in ref. 62. The wild-type AB strain, the *chd1l*[sa14029] (TL) mutant line (#15474), carrying the mutation C > T at the genomic location Chr.6:36844273 (GRCz11), the *Tg(olig2:EGFP)vu12* (AB) (#15211) line were obtained from the European Zebrafish Resource Center. All fish lines reproduce normally, no skewed sex ratio was observed and *chd1l* homozygote mutants were recovered in the expected Mendelian ratio. Genotyping of the *chd1l*[sa14029] mutant line was performed by a PCR following restriction digestion. Wholemount immunostaining involved specific protocols for fixation, permeabilization, blocking, and antibody incubation, followed by imaging using MacroFluo ORCA Flash (Leica) system. Additionally, imaging procedures for the oligodendrocyte lineage, motor neurons, and acetylated tubulin staining were performed on fixed larvae to visualize cellular structures and assess specific cellular populations. For the statistical analysis, at least 3 replicates for each genotype were analyzed and the number of larvae/replicates is indicated. A detailed protocol is available in supporting information.

**Human iPSC *CHD1L* knock-down experiments**
In this study, we used iPSCs derived from three subjects. Fibroblasts were converted to iPSCs using mRNA reprogramming as previously described[63,64], and in accordance with IRB approval from the Ethical Review Boards in Stockholm, Sweden. iPSC lines were expanded on Matrigel-coated plates and purified using Anti-TRA-1-60 MicroBeads via magnetic cell sorting (MACS). NPCs were generated through dual SMAD pathway inhibition, as reported earlier[65]. Briefly, iPSCs were cultured in V-bottom ultra-low-attachment 96-well plates for 1 week in embryoid body medium, then transferred to Matrigel-coated 6-well plates to induce neural rosettes. After manual isolation, rosettes were expanded on Matrigel-coated plates in NPC media. NPCs were purified through CD271 negative selection followed by CD133 positive selection via MACS. For neuronal differentiation, NPCs were cultured on poly-L-ornithine and laminin-coated plates. Differentiation proceeded over

5 weeks with media changes and supplements as described in supporting material. *CHD1L* knock-down was achieved using Accell human *CHD1L* siRNA SMARTpool. Immunocytochemistry involved fixing cells, blocking with a solution of BSA and Triton X-100, and incubating with primary antibodies overnight. Secondary antibodies were applied before counterstaining with DAPI. Imaging was performed using confocal microscopy and analyzed with the ImageJ software. Calcium imaging in 5-week-old neuronal cultures was performed using the Cal-520® AM dye. Spontaneous calcium activity was recorded using a Nikon CrEST X-Light V3 Spinning Disk microscope. The analysis involved defining regions of interest (ROIs) and calculating fluorescence changes over time. A detailed protocol is available in supporting information.

**Reporting summary**
Further information on research design is available in the Nature Portfolio Reporting Summary linked to this article.

## Data availability
The 450K array methylome data generated from whole blood (cohort 1) are available in the Gene Expression Omnibus (GEO) database under accession number GSE106648. The RNA-sequencing data used for correlation network analysis in MS brain are accessible under the accession numbers GSE174647, GSE118257, GSE179427, PRJNA544731 with details presented in Supplementary Data 11. Results from the genetic association analysis (cohort 3) are available in Supplementary Data. The personal information including genetic data is available under restricted access for the GDPR regulations, access can be obtained under a Data Access Agreement. Source data are provided with this paper.

## Code availability
The codes for data analysis used in this study are available at https://gitlab.com/jagodiclab/pahlevan_ppms_chd1l.

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

## Acknowledgements

This study was supported by grants from the Swedish Research Council, the Swedish Association for Persons with Neurological Disabilities, the Swedish Brain Foundation, the Swedish MS Foundation, the Stockholm County Council - ALF project, the European Union's Horizon 2020 research, innovation programme (grant agreement No 733161), the European Research Council grant (grant agreement No 818170), the Knut and Alice Wallenberg Foundation, Hedlund Foundation, Erling Persson Foundation, Åke Wiberg Foundation, and Karolinska Institute's funds. The Zebrafish work was funded by Agence Nationale de la Recherche under the projects JCJC ANR-17-CE12-0006 and ANR-22-CE12-0011 (C.G.). The Zebrafish work of the Inter-disciplinary Thematic Institute IMCBio, as part of the ITI 2021-2028 program of the University of Strasbourg, CNRS, and Inserm, was supported by IdEx Unistra (ANR-10-IDEX-0002), and by SFRI-STRAT'US project (ANR 20-SFRI-0012) and EUR IMCBio (ANR-17-EURE-0023) under the framework of the French Investments for the Future Program. L.K. and P.S. are supported by a fellowship from the Margaretha af Ugglas Foundation. M.P.K. was supported by McDonald Fellowship from Multiple Sclerosis International Federation (MSIF) and has received funding from the Innovative Medicines Initiative 2 Joint Undertaking (JU) under grant agreement No 875510. The JU receives support from the European Union's Horizon 2020 research and innovation programme and EFPIA and Ontario Institute for Cancer Research, Royal Institution for the Advancement of Learning McGill University, Kungliga Tekniska Hoegskolan, Diamond Light Source Limited. This communication reflects the views of the authors and the JU is not liable for any use that may be made of the information contained herein. C.S.C. is supported by the Blanceflor Boncompagni Ludovisi, née Bildt Foundation. P.C. is supported by NIH R35-NS111604. C.G. is a permanent INSERM investigator. A.H. is supported by a FRQS Clinical Research Scholarship (349722). M.V.L. is supported by an IMCBio PhD fellowship. The funders of the study had no role in study design, sample acquisition, data collection, data analysis, data interpretation, or writing of the manuscript. We are grateful to all volunteers for contributing to the study. We thank Maria Kalomoiri, Jessica Aguilar, and Jacqueline Hammer for their help in the experimental setup, Dr. Ewoud Ewing for statistics assistance, Dr. Alexander Espinosa for providing the plasmid construct and Prof. Lucas Schirmer for providing the CUX2⁺ data. We thank the IGBMC Imaging Center and the IGBMC Zebrafish Facility, in particular Sandrine Geschier. We are also grateful to Anastasiya Petrova for her technical assistance. We acknowledge the KIGene facility at the Center for Molecular Medicine and Karolinska Institutet for Sanger sequencing service. The computations were enabled by resources provided by the Swedish National Infrastructure for Computing (SNIC) through the Uppsala Multi-disciplinary Center for Advanced Computational Science (UPPMAX), partially funded by the Swedish Research Council through grant agreement no. 2018-05973. The authors acknowledge support from the National Genomics Infrastructure in Stockholm funded by Science for Life Laboratory, the Knut and Alice Wallenberg Foundation, and the Swedish Research Council (through grant agreement no. 2018-05973). We thank deCODE for the genotyping of the Swedish population (https://www.decode.com/). We acknowledge support from Servier Medical Art (https://smart.servier.com) and Biorender for generating schematics in Figs. 4 and 6, respectively.

## Author contributions

M.P.K. designed and conducted the cloning, sequencing, and PCR analyses, in vitro DNA methylation assays, dCas9-TET1 experiments, performed statistical analyses, and interpreted the data, with help from C.S.C. S.B. and F.W. assisted in lentiviral production and in vitro experiments. Y.L. performed genome-wide DNA methylation and meQTL analyses (cohort 1). A.Gy. carried out locus-specific meQTL (cohort 2). A.Gi., P.S., K.S., and A.Gy. conducted genetic association analyses (cohort 3). A.H. performed phasing and imputation of genotypes. T.V.S.B. performed correlation network analysis and M.G. supervised network analysis. T.J. aided in data acquisition. M.V.L. and C.G. designed and performed the zebrafish experiments. S.S., C.M.S., and A.G. designed and conducted the iPSC experiments. I.K. and F.E. provided genetic data. P.C. provided brain sequencing data. M.S. contributed to the genotyping of the Italian cohort. M.F., M.S., F.E., A.Gi., T.O., J.H., and P.C. contributed to the collection of patient samples. L.K. supervised the study, performed analyses, interpreted the data, and wrote the manuscript with the contribution from M.P.K. and M.J., and all co-authors. M.J. conceptualized, designed, interpreted, and supervised the study. All authors read and approved the final manuscript.

## Funding

## Competing interests

The authors declare no competing interests.

## Additional information

[1]Department of Clinical Neuroscience, Karolinska Institutet, Center for Molecular Medicine, Karolinska University Hospital, Stockholm, Sweden. [2]Neurology and Neurorehabilitation Units, IRCCS San Raffaele Hospital, Milan, Italy. [3]Laboratory of Human Genetics of Neurological Disorders, Division of Neuroscience, IRCCS San Raffaele Scientific Institute, Milan, Italy. [4]Università Vita-Salute San Raffaele, Milan, Italy. [5]Center for Neurology, Academic Specialist Center, Stockholm, Sweden. [6]Department of Bioinformatics, Institute for Physics chemistry and Biology (IFM), Linköping university, Linköping, Sweden. [7]Department of Physiology and Pharmacology, Karolinska Institutet, Stockholm, Sweden. [8]Institut de Génétique et de Biologie Moléculaire et Cellulaire, Illkirch, France. [9]Centre National de la Recherche Scientifique, UMR7104, Illkirch, France. [10]Institut National de la Santé et de la Recherche Médicale, U1258, Illkirch, France. [11]Université de Strasbourg, Strasbourg, France. [12]The Neuro (Montreal Neurological Institute-Hospital), Montréal, QC, Canada. [13]Department of Neurology and Neurosurgery, McGill University, Montréal, QC, Canada. [14]Department of Human Genetics, McGill University, Montréal, QC, Canada. [15]MOE Key Laboratory of Metabolism and Molecular Medicine, Department of Biochemistry and Molecular Biology, School of Basic Medical Sciences and Shanghai Xuhui Central Hospital, Fudan University, Shanghai, China. [16]Department of Medicine, Solna, Karolinska Institutet, Center for Molecular Medicine, Karolinska University Hospital, Stockholm, Sweden. [17]Neurophysiology Unit, IRCCS San Raffaele Hospital, Milan, Italy. [18]Neuroimaging Research Unit, Division of Neuroscience, San Raffaele Scientific Institute, Milan, Italy. [19]Department of Neurology, Icahn School of Medicine at Mount Sinai, New York, USA. [20]Center for Psychiatry Research, Department of Clinical Neuroscience, Karolinska Institutet, Stockholm, Sweden. [21]Stockholm Health Care Services, Stockholm County Council, Stockholm, Sweden. [22]These authors contributed equally: Antonino Giordano, Chiara Starvaggi Cucuzza, Tejaswi Venkata S. Badam, Samudyata Samudyata, Marianne Victoria Lemée. [23]These authors jointly supervised this work: Lara Kular, Maja Jagodic. ✉e-mail: lara.kular@ki.se; maja.jagodic@ki.se

