## [Peer Review File · Nature Communications]

A genetic-epigenetic interplay at 1q21.1 locus underlies *CHD1L*-mediated vulnerability to Primary Progressive Multiple SclerosisREVIEWER COMMENTS

Reviewer #1 (Remarks to the Author):

In this interesting study, the authors have found that hypermethylation of the 1q21.1 locus gene (CDH1L, PRAKAB2, and FM05) is associated with increased risk for progressive multiple sclerosis (PPMS). This chromosomal locus contains human-specific genes and has been shown to be associated with neurodevelopment. Mutations at this locus have been shown to increase the risk of a range of brain developmental disorders. The study uses two cohorts and samples collected from the brain and blood. The findings from the study will widen our understanding of the pathogenesis of PPMS. Several points that need to be addressed are listed below, and I hope those comments will be helpful.

1. What are the major brain cell types that are impacted by hypermethylation in the 1q21.1 region? Although neurons have roles to play, it will be interesting to see if the oligodendroglia lineage-specific transcription factor binding regions (Sox10, MYRF1, etc.) are also hypermethylated. This would help in understanding the role of the myelin regeneration process in the context of PPMS pathogenesis.
2. I am not very convinced by the choice of neuronal-like cells for functional assays. Further, it does not address how the hypermethylation of loci like CDH1L is likely to impact the function and physiology of these cells. Additional experiments are needed to clarify if the hypermethylation leads to neuronal deficits.
3. Network analysis implicating the role of CDH1L and PRAKAB2 in PPMS brain pathology is quite interesting to understand the specific disease mechanisms associated with PPMS. Can these two genes, in combination or when deleted or duplicated, still be associated with PPMS brain pathology?
4. I am curious if the gender and age of PPMS patients influence the methylation status of the 1q21.1 locus.
5. Did the author also see any changes in the methylation status of the neighbouring TAR region in PPMS patients in addition to the critical region of 1q21.1?
6. Please discuss what factors are likely to contribute to the increased methylation status of the 1q21.1 region in PPMS patients.

Reviewer #2 (Remarks to the Author):

This study aimed to identify differentially methylated regions associated with primary progressive MS. The study is warranted based on the current lack of knowledge of disease pathology underpinning progressive MS and the ultimate need for better diagnostics and treatments to prevent this eventual outcome in many MS patients.

The research design although quite complex incorporates multiple independent comparative cohorts (RRMS, SPMS, PPMS and HC), multiple cell types (blood, neurobiological) and multiple layers of omic data (genotype, methylation, transcript levels).

Based on the initial EWAS a hypomethylated region on chr 1q21 was convincingly shown to be associated with PPMS. This locus was shown to be under genetic control using blood and neuro meQTL analysis.

Since the meQTL is intergenic the investigators rightly tested for trans-acting effects on gene expression in neuron-like cells incorporating gene editing technology. This nice approach highlighted the wide-spread regulatory effects of the meQTL in these cells.

Finally, a bioinformatic-based gene network analysis using publically-available brain data and data from MS patients specifically implicated several genes (and modules) involved in PPMS.

Overall, this is a comprehensive study incorporating multiple layers of evidence to causally

implicate several genes in neuronal pathology of PPMS. Despite the complex design (workflow) the manuscript is well constructed and written and the conclusions are supported by the results. Overall, this an important and novel piece of research.

One area I think needs addressing is the following. The statistical aspects of the several results sections are very probability (p-value) focused. Whilst this is of course important for interpretation I would like to see some measures of effect size and direction mentioned in the text ie. to accompany the p-values. For example, what is the delta-beta (range, average, max) of the hypomethylation locus on 1q21? The meQTL is under genetic control but what is the strength (heritability) of this? Can you quantify. What are the odds ratios for the meQTL SNPs in the large PPMS cohort association study. All these effects are important to help gauge the relative importance of the finding in the complex context of PPMS pathology.

Reviewer #3 (Remarks to the Author):

In this manuscript, the authors explore the molecular mechanism of PPMS through multiple cohorts, cross-tissue, and multilayered data. The results indicated that hypermethylation at 1q21.1 was associated with low expression of CHD1L and PRKAB2 and high risk of PPMS. The manuscript has a large amount of data, complicated statistical work and novel results. However, the manuscript is poorly read, with errors in numerical details and a lack of depth in the discussion of the novel results. Careful adjustment of this manuscript is recommended.

1. Personally, I don't think this title can cover all the results of this paper.
2. Please check whether multiple sclerosis is a neurodegenerative disease. I couldn't find any definition of neurodegenerative disease in the first reference (Kutzelnigg, A. et al Cortical demyelination and diffuse white matter injury in multiple sclerosis. Brain).
3. The authors used at least 5 cohorts to investigate the methylation of the PPMS-associated 1q21.1 locus, but didn't describe the details and reasons for using of all the cohorts. They were messy and wasn't closely related. Why didn't use cohort 1 and cohort 2 but cohort 3 to proceed association analysis? Is the brain tissue used to locus-specific meQTL and correlation network analysis from the same cohort? In the second part of result, the authors mentioned 'only eight out of the 19 meQTL-SNPs identified in cohort 1 could be assessed in this cohort (cohort 2)', would some important information be left out? Was it limited by technology or the quality of blood samples or something else?
4. The authors didn't discuss why locus-specific demethylation at CpGs of the identified DMR at the 1q21.1 DMR did not affect any significant gene expression changes in HEK293T cells? And why 'the expression of FMO5 and other proximal transcripts, PDIA3P1 and PFN1P5, did not vary significantly'.
5. Can the authors explain the correlation among the expression level of PRKAB2, the methylation level and the PPMS risk? (line 197-200)
6. Please add the legends of Figure 3D. What does the variant ID represent? It is difficult to read.
7. Line 249, I couldn't find the related information in the figure 3C. Please label the correct figure.
8. In the fifth section of result, the authors indicated ' but CHD1L, FMO5 and PRKAB2 genes belonged to significantly co-expressed genes networks only in the cortex and tibial nerve', but the typical pathology of MS is focal plaques of primary demyelination. However, the authors didn't discuss the different location of the pathology and genes co-expression in the part of discussion.
9. To validate the bioinformatics data, the authors used SH-SY5Y cell line and HEK293T cell line, which are mostly used for modeling neurodegenerative diseases. Are they the recognized cell model for MS?
10. Please confirm the details of the data in the article. For example, 140 HC in cohort 1 is shown in Figure 1. According to the description in the manuscript, it seems that the number of HC should be 139.

Point-by-point response to reviewers' comments

Reviewer #1 (Remarks to the Author):

Comments: In this interesting study, the authors have found that hypermethylation of the 1q21.1 locus gene (CDH1L, PRAKAB2, and FM05) is associated with increased risk for progressive multiple sclerosis (PPMS). This chromosomal locus contains human-specific genes and has been shown to be associated with neurodevelopment. Mutations at this locus have been shown to increase the risk of a range of brain developmental disorders. The study uses two cohorts and samples collected from the brain and blood. The findings from the study will widen our understanding of the pathogenesis of PPMS. Several points that need to be addressed are listed below, and I hope those comments will be helpful.

Response:

We would like to thank the Reviewer for thorough scrutiny of our manuscript and positive evaluation of the findings. We appreciate Reviewer's constructive comments and suggestions that have significantly improved our study and interpretations.

1. What are the major brain cell types that are impacted by hypermethylation in the 1q21.1 region? Although neurons have roles to play, it will be interesting to see if the oligodendroglia lineage-specific transcription factor binding regions (Sox10, MYRF1, etc.) are also hypermethylated. This would help in understanding the role of the myelin regeneration process in the context of PPMS pathogenesis.

Response to Comment #1:

We appreciate this relevant comment that we have addressed as follows:

- 1) As the regulatory effect of the 1q21.1 DMR region is methylation-dependent, we asked whether the basal methylation level of the DMR in different CNS cell types could be indicative of lineage-specificity. We examined methylation levels at the DMR-CpGs in neuronal (NeuN+, n = 34) and non-neuronal/glia (NeuN-, n = 56) nuclei (new Supplementary Fig. 2a) sorted from *postmortem* brain tissue as well as in individual sorted (*ex vivo*) or enriched (*in vitro*) CNS cell types (neurons, oligodendrocytes and their progenitors, astrocytes and microglia, n = 4-7 samples/cell type, new Supplementary Fig. 2b). There was no noticeable difference between methylation levels in different CNS cell types, all exhibiting relatively low levels of methylation. This indicates that the 1q21.1 region is not constitutively repressed by DNA methylation and has a potential to exert regulatory effect in different CNS cell types. However, since genetic information was not available for these samples, one cannot exclude the possibility that the identified PPMS-specific SNPs, governing the methylation patterns in 1q21.1, display cell type-specific effects.
- 2) As suggested by the Reviewer, we interrogated the sequence at 1q21.1 DMR region for the presence of transcription factor (TF) binding sites using the JASPAR CORE database (summarized in new Supplementary Fig. 2c). Among many TF binding sites, we found binding sites of multiple TFs known to be involved primarily in development of neurons (NEUROD1¹, ATOH1², ASCL1³, NEUROG1/2, TBR1), together with binding sites for TF also relevant for oligodendrocytes (SOX10⁴, ASCL1⁵), and astrocytes (NFIB⁶) as well. This suggests a potential role of 1q21.1 regulatory region in multiple CNS cell types, and particularly neuronal lineage. Of note, none of these TF binding sequences encompass CpG sites.

Notably, while addressing the functional effect of the locus (*details are provided in response to Comment #3*), we obtained evidence that *CHD1L* is relevant for neuronal functioning in humans and neuronal development in zebrafish. Undoubtedly, while we mainly focus on the role of *CHD1L* in the neuronal lineage, we cannot exclude the possibility that *CHD1L* may also affect glial cells, as suggested by the decreased number of oligodendrocyte precursors in the developing zebrafish.

We have now added these new analyses in the revised manuscript as new **Supplementary Fig. 2**. The corresponding text in the revised **Results** section (page 7) reads as follows:

These data connecting genetic variation, methylation, and transcription in the extended 1q21.1 region are consistent with genomic annotations of the DMR underscoring putative regulatory features as well as short-range physical interaction with proximal chromatin segments harboring eQTL genes (Supplementary Fig. 1). **The analyses of DNA methylation levels and transcription factor (TF) binding**

motifs in 1q21.1 suggest that the region has a potential to exert regulatory functions in all CNS cell types, particularly neurons (Supplementary Fig. 2).

Supplementary Fig. 2. Methylation levels across brain cells and transcription factor binding sites in the 1q21.1 region. **a.** Methylation levels across CpGs in 1q21.1 region measured by Illumina arrays in **la.** Sorted NeuN+ (neuronal, $n = 34$) and NeuN- (non-neuronal/glia, $n = 56$) nuclei from *postmortem* brain tissue (data from Kular *et al.*^{2,3}) and **b.** Sorted and/or enriched brain cell types. Neurons ($n = 3$) comprise *ex vivo* sorted nuclei, oligodendrocytes (OL, $n = 7$) and microglia ($n = 5$) contain both *in vitro* expanded and *ex vivo* sorted samples, while oligodendrocyte progenitor cells (OPC, $n = 4$) and astrocytes ($n = 4$) are exclusively derived from *in vitro* primary cultures (genomic DNA purchased from Celprogen). **c.** JASPAR sequence analysis revealed binding sites for several transcription factors (TF) important for neuronal (blue), oligodendrocytic (red) and astrocytic (black) specification. The green colored CpGs indicate the hypermethylated CpGs found in PPMS, relative to RRMS and SPMS patients.

And the corresponding methods are added to the revised **Methods** section (page 19):

TF binding evaluation

We utilized the JASPAR core database⁵² through the UCSC Genome Browser track data hub to identify putative TFs with binding sites in the 1q21.1 DMR region, using default criteria. We then manually annotated and filtered for TFs with known functions in development of neuronal and glial cells.

Methylation levels across brain cells

We investigated the methylation levels of the 1q21.1 DMR-CpGs in neuronal and non-neuronal cell types by extracting previously published Illumina array-derived data (described in detail in Kular *et al.*^{10,12}) generated by sorting neuronal (NeuN+, $n = 34$) and non-neuronal/glia (NeuN-, $n = 56$) nuclei from postmortem human brain samples. Additionally, individual brain cell types were sorted and/or enriched using primary *in vitro* cultures (purchased from Celprogen, catalog #36058DNA, 37089DNA, 36055DNA and 736055-22DNA) include *ex vivo* sorted neuronal nuclei ($n = 3$), *in vitro* and *ex vivo* samples of oligodendrocytes (OL, $n = 7$) and microglia ($n = 5$), and exclusively *in vitro* cultured oligodendrocyte

progenitor cells (OPC, n = 4) and astrocytes (n = 4), with methylation data available in GEO database under the accession number GSE166207.

2. I am not very convinced by the choice of neuronal-like cells for functional assays. Further, it does not address how the hypermethylation of loci like CDH1L is likely to impact the function and physiology of these cells. Additional experiments are needed to clarify if the hypermethylation leads to neuronal deficits.

Response to Comment #2:

We appreciate this relevant comment regarding the link between the genes regulated by the methylation of the locus with neuronal development and functionality. To address the functional impact of *CHD1L* on the CNS and neurons in particular, we have performed extensive *in vitro* (in human iPSC-derived neurons) and *in vivo* (zebrafish) experiments (new Fig. 6, described in **Comment #3**). As detailed in response to the following **Comment #3**, these data jointly demonstrate that loss of *CHD1L* influences the development and function of neurons.

The choice of the cell lines for various types of experiments has also been clarified (*details are provided in response to Comment #4 from Reviewer #3*).

3. Network analysis implicating the role of CHD1L and PRKAB2 in PPMS brain pathology is quite interesting to understand the specific disease mechanisms associated with PPMS. Can these two genes, in combination or when deleted or duplicated, still be associated with PPMS brain pathology?

Response to Comment #3:

We thank the Reviewer for this insightful comment and suggestion, and we appreciate the opportunity to present additional experimental data that shed light on the functional implications of *CHD1L* and its impact on cellular physiology.

As we mentioned in the discussion, the 1q21.1 locus displays a very complex genomic architecture and is associated with multiple brain disorders. Moreover, we noted that some of the SNPs conferring risk for PPMS are also associated with Schizophrenia⁷.

Previous studies have reported functional effects of 1q21.1 locus (encompasses *CHD1L*, *PRKAB2*, *FMO5*, *NBPF12* genes) deletion or duplication on neuronal function. As cited in the discussion, patient-derived iPSC-generated neurons with 1q21.1 deletion or duplication exhibit opposite effects on various characteristics, such as proliferation, differentiation potential, neuronal maturation, synaptic density and functional activity (Chapman *et al.*⁸). Similarly, reciprocal 1q21.1 CRISPR/Cas9-induced CNVs in isogenic hES cell lines have a significant impact on the cell fate during early neurodevelopment (this work by Nomura *et al.*⁹ is now cited in the revised manuscript). Moreover, the number of copies of the 1q21.1 distal region influenced brain structures and resulted in decreased cognitive performance in individuals, as compared to those without these genetic variations (this work by ENIGMA-CAN¹⁰ is now cited in the revised manuscript). While these studies strongly suggest the involvement of the 1q21.1 region in neuronal development and function, they do not directly address the impact of individual genes within the genomic locus.

Our data also strongly support the role of *CHD1L*:

- Our reporter and CRISPR experiments demonstrate that methylation at the 1q21.1 DMR regulated *CHD1L* gene expression (Fig. 4) and concordant meQTL/eQTL effect was observed for *CHD1L* within the CNS tissue, as opposed to *PRKAB2* (Fig. 3d).

Figure 4

Figure 3d

Revised Fig. 3 panel legend: d. Association between genetic variation, DNA methylation (using xQTL serve platform¹¹) and gene expression (using GTEx database) in the extended locus for SNPs tagging four variants (LD blocks) displaying significant association with PPMS in the meta-analysis of Swedish (SWE) and Italian (ITA) cohorts. Protective and risk genetic effects are depicted in green and orange, respectively. Green and orange colors reflect protective and risk effect on PPMS risk conferred by the genetic variants. Blue and red colors represent negative and positive effect of the minor allele, respectively, on methylation or expression. The data shown in this panel are available in Supplementary Data 7 (meQTL, eQTL) and 8 (genetic analysis).

- The gene ontology analysis of the *CHD1L* network in the PPMS postmortem brain revealed functional annotations related to neuronal and neurodegenerative processes, compared to *PRKAB2* (Fig. 5c).
- Notably, we detected a significant overlap of the *CHD1L* gene network (but not *PRKAB2* and *FMO5* networks), with the gene signature of CUX2⁺ excitatory neurons, previously reported to be specifically affected in patients with MS (Schirmer et al ¹²), with the closest interactors of *CHD1L* (but not *PRKAB2* and *FMO5*) in PPMS also replicated in CUX2⁺ excitatory neurons.
- Moreover, one study links *CHD1L* with early neuroepithelial differentiation of hESCs¹³ (this work is now cited in the revised manuscript).

Given these lines of compelling evidence supporting the role of *CHD1L* in the context of PPMS, we conducted functional experiments using *CHD1L*-targeted knock-down *in vivo* in the zebrafish and *in vitro* human iPSC-derived neurons. Our approach and findings are summarized below:

- (1) In *chd1l* knock-out model in zebrafish, we found a significant reduction in the number of axonal tracts in the brain of *chd1l* mutant (*chd1l*^{+/-}) zebrafish larvae compared to controls (*chd1l*^{+/+}), supporting a role of *CHDL1* in axonal development. This role was specific for the CNS in the zebrafish, as defective sprouting or pathfinding deficits were not as evident in peripheral neurons. Moreover, by crossing the *chd1l* mutant (*chd1l*^{+/-}) and control lines with the Tg(olig2:EGFP) reporter line, we also detected a notable decrease in the number of oligodendrocyte progenitors and developing motor neurons in the brain and spinal cord of the mutant larvae, compared to controls, thereby underscoring a perturbation in both neuronal and oligodendrocyte lineage. Overall, these findings suggest a role for *chd1l* contributes to the CNS development and integrity also *in vivo*. Although a parallel between zebrafish and human physiopathology should be taken with caution, these results provide valuable insights into the functional consequences of *chd1l* gene and suggest that lower levels of *CHD1L* consequent to hypermethylation could modulate neuronal function in a manner that may contribute to deficits observed in PPMS patients.
- (2) In human iPSC-derived neurons in which *CHD1L* was silenced from the neural progenitor cell (NPC) to differentiated forebrain neurons stage, while *CHD1L* knock-down did not affect the expression of neuronal differentiation markers, *CHD1L*-deficient differentiated neurons manifested structural (smaller neurites) and functional (lower amplitude of Ca²⁺ dynamics) abnormalities indicating defective branching and neuronal activity.

We believe the novel *in vitro* and *in vivo* findings significantly enhance the understanding of the functional repercussions of *CHD1L* dysregulation. In addition, even though we address the role of *CHD1L* in the neuronal lineage, we cannot exclude the possibility that additional cell types may be affected or that additional genes within the 1q21.1 locus may also play a role.

The new data are now presented in our revised manuscript as a **Result** section (page 11) with the corresponding **Figure 6** and **Supplementary Fig. 5**, and reads as follows:

End of the section “***CHD1L gene network is enriched in PPMS brain tissue***”:

Thus, we detected a significant overlap between PPMS-associated *CHD1L* gene network and the gene signature of neurons previously reported to be specifically affected in MS²⁴, which, together with the concordant meQTL and eQTL effects of PPMS-associated variants for this gene, implicate *CHD1L* gene in neuronal vulnerability in PPMS.

Dysregulation of CHD1L affects neuronal development and activity

Previous studies unveiled the essential role of *CHD1L* at early embryonic developmental stages, with its absence at the zygote-stage causing early developmental arrest in rodents^{25,26}. To investigate the role of *CHD1L* in cellular functionality relevant for MS pathology *in vivo*, we utilized a *chd1l* mutant zebrafish line and examined whether dosage-dependent *chd1l* expression affected cells of the neuroglial lineage in the head and the caudal part of the zebrafish larvae (**Fig. 6a**). Three days post-fertilization (dpf), *chd1l*^{+/-} mutant larvae displayed a significant reduction in the number of axonal tracts projecting from the optic tecta compared to control *chd1l*^{+/+} larvae (**Fig. 6b**, mean axonal tracts = 12.2 vs. 10.7 respectively, $p = 8.7 \times 10^{-6}$). Of note, mutant larvae did not exhibit significant alteration of sprouting and/or pathfinding of peripheral neurons (Supplementary Fig. 5). We next used the Tg(olig2:EGFP) reporter line that marks oligodendrocyte precursors and motor neurons at 3 dpf^{27,28} and found a significant decrease of the number of olig2-expressing cells in the hindbrain (**Fig. 6c**, normalized

mean = 0.19 vs. 0.16 in *chd1l*^{+/+} and *chd1l*^{+/-} respectively, $p = 4.1 \times 10^{-6}$) and dorsally migrating from the spinal cord (**Fig. 6d**, normalized mean = 36.31 vs. 29.42 in *chd1l*^{+/+} and *chd1l*^{+/-} respectively, $p = 8.1 \times 10^{-3}$) of *chd1l* mutant larvae. These findings indicate that *chd1l* contributes to the CNS development and integrity *in vivo*.

To further examine the role of *CHD1L* in neuronal function specifically, we used iPSC-derived neural progenitor cells (NPCs) patterned via dual SMAD inhibition and differentiated them into forebrain neurons for five weeks (DIV 35) in the presence of *CHD1L* siRNAs (knock-down, KD) or non-targeting (NT) siRNAs (**Fig. 6e**). The NPCs expressed markers of early neuroectodermal induction (PAX6, SOX1) and following differentiation, neurons exhibited MAP2⁺ neurites marked by putative synapses (SYN1) (**Fig. 6f**). The successful knock-down of *CHD1L* during differentiation did not affect the expression of intermediate progenitor and migration markers (*EOMES*, *NCAM1*) or cortical layer markers (*CTIP2*, *SATB2*) (**Fig. 6g**). Instead, *CHD1L*-deficient neurons presented branching abnormalities reflected by smaller neuritic protrusions, while the synaptic density seemed unaffected (**Fig. 6h**). *CHD1L*-knockdown also resulted in neuronal functional impairment detectable as reduced calcium intensity signal, overall suggesting weaker electrical activity in neurons with lower levels of *CHD1L* (**Fig. 6i**).

Altogether, our data imply that dysregulation of *CHD1L* in CNS neurons may lead to functional neuronal impairment.

Figure 6

Figure 6. Functional impact of low *CHD1L* expression in neurons and oligodendrocytes. a. Schematic of the experimental design in the zebrafish. **b.** Dorsal view of control and *chd1l* ^{+/+}/_{-/-} larvae at 3 dpf stained with acetylated tubulin. Barplot of the inter-tecta axonal tract projections count after acetylated tubulin staining of 3 dpf control and *chd1l* ^{+/+}/_{-/-} larvae (mean ± SEM for n = 3 replicate per genotype with 30 larvae/replicate). **c.** Dorsal view of Tg(olig2:EGFP); *chd1l* ^{+/+}/_{-/-} and Tg(olig2:EGFP); *chd1l* ^{+/+}/_{-/-} at 3 dpf with the hindbrain used for olig2-positive cells count outlined in yellow. Barplot of the number of olig2-positive cells per μm² in the hindbrain of Tg(olig2:EGFP); *chd1l* ^{+/+}/_{-/-} and Tg(olig2:EGFP); *chd1l* ^{+/+}/_{-/-} larvae at 3 dpf (mean ± SEM of n = 3 per genotype and 12-20 larvae/replicate). **d.** Lateral view of the spinal cord of Tg(olig2:EGFP); *chd1l* ^{+/+}/_{-/-} and Tg(olig2:EGFP); *chd1l* ^{+/+}/_{-/-} larvae at 3 dpf. Barplot of the number of olig2-positive cells migrating dorsally from the spinal cord of Tg(olig2:EGFP); *chd1l* ^{+/+}/_{-/-} and Tg(olig2:EGFP); *chd1l* ^{+/+}/_{-/-} larvae at 3 dpf (mean ± SEM of n = 3 replicate per genotype and 15-20 larvae/replicate). **e.** Schematics showing *in vitro* differentiation of iPSCs into forebrain neurons. **f.** Representative confocal images of markers of neuroectodermal NPCs (PAX6, SOX1) differentiated neurons (MAP2, SYN1). **g.** qPCR data of *CHD1L*, *EOMES*, *SATB2* and *CTIP2* transcripts in non-

targeting (NT) and *CHD1L* knock-down (KD) neurons. Data points correspond to experimental duplicates in three independent subject lines (highlighted in different colors). **h.** Quantification of neurite length (μm) and synaptic density in NT and KD neurons. **i.** Quantification of background-corrected changes in calcium-sensitive dye fluorescent intensity in NT and *CHD1L* KD neurons. A representative confocal image is shown. Data points correspond to averaged active regions (ROIs, 10 cells) per field of view. Five to six fields of view were included for each subject line and condition. Statistical significance was assessed using the Wilcoxon test (b, c,), Student's T-test (d), Mann-Whitney U test (g, h, i) for p-value. A, anterior; P, posterior; dpf, days post-fertilization; NT, non-targeting siRNA; KD, *CHD1L*-targeting siRNA.

Supplementary Fig 5.

Supplementary Fig. 5. Effect of *chd1l* loss on peripheral axonal abnormalities in 3 dpf zebrafish larvae. **a.** Schematic of the experimental design. **b.** Lateral view of control *chd1l*^{+/+} and *chd1l*^{+/-} larvae stained with acetylated tubulin at 3 dpf. Scale bar: 15 μm . **c.** Examples of axonal projections: normal branching (top), ectopic branching (middle) and reduced/absent branching (bottom) used for qualitative quantification of peripheral axons. **d.** Stacked barplot of the percentage of larvae with axonal projection defects in peripheral neurons. Number of abnormal projections were counted across eight metamers in control and *chd1l*^{+/-} larvae. Data are expressed as percentage of larvae for N = 3 replicates for each genotype. Fischer's exact test is performed for p-value. A, antero; P, posterior, dpf: days post-fertilization.

The corresponding text in the revised **Discussion** (page 16) reads as follows:

Findings from our study converge to several lines of evidence, namely the concordant meQTL/eQTL effect in the CNS, methylation-mediated *CHD1L* regulation and detection of dysregulated *CHD1L*-gene network pertaining to neuronal processes the PPMS brain, implicating *CHD1L* in neuronal vulnerability in PPMS. To formally address the role of *CHD1L* in neuronal function, we conducted functional experiments using *CHD1L*-targeted knock-down *in vivo* in the zebrafish and *in vitro* human iPSC-derived neurons. While specific abrogation of *chd1l* in rodents is lethal at the zygote stage and impairs neuroectodermal development at early embryonic stage²⁶, the data from our study further delineate a specific effect of *chd1l* on the axonal tract, neurite projection and neuronal activity *in vitro* and *in vivo*. Such effect aligns with the notion that neuro-axonal damage and ensuing network dysconnectivity contribute to neurodegeneration in MS^{46,47}, with greater axonal loss being observed in the PPMS brain compared to SPMS⁴⁸. Yet, *CHD1L* is likely to influence the functionality of other cell types involved in the CNS development, such as oligodendrocyte lineage as suggested by results in the zebrafish, and one cannot exclude additional influences of other genes, such as *PRKAB2*, from the 1q21.1 locus⁴⁵.

The corresponding text in the revised **Methods** (page 23) reads as follows:

Human iPSC *CHD1L* knock-down experiments

In this study, we used iPSCs derived from three subjects. Fibroblasts were converted to iPSCs using mRNA reprogramming as previously described^{62,63}, and in accordance with an IRB approval from the Ethical Review Boards in Stockholm, Sweden. iPSC lines were expanded on Matrigel-coated plates and purified using Anti-TRA-1-60 MicroBeads via magnetic cell sorting (MACS). NPCs were generated through dual SMAD pathway inhibition, as reported earlier⁶⁴. Briefly, iPSCs were cultured in V-bottom

ultra-low-attachment 96-well plates for 1 week in embryoid body medium, then transferred to Matrigel-coated 6-well plates to induce neural rosettes. After manual isolation, rosettes were expanded on Matrigel-coated plates in NPC media. NPCs were purified through CD271 negative selection followed by CD133 positive selection via MACS. For neuronal differentiation, NPCs were cultured on poly-L-ornithine and laminin-coated plates. Differentiation proceeded over 5 weeks with media changes and supplements as described in supporting material. *CHD1L* knock-down was achieved using Accell human *CHD1L* siRNA SMARTpool. Immunocytochemistry involved fixing cells, blocking with a solution of BSA and Triton X-100, and incubating with primary antibodies overnight. Secondary antibodies were applied before counterstaining with DAPI. Imaging was performed using confocal microscopy and analyzed with the ImageJ software. Calcium imaging in 5-week-old neuronal cultures were performed using the Cal-520® AM dye. Spontaneous calcium activity was recorded using a Nikon CrEST X-Light V3 Spinning Disk microscope. Analysis involved defining regions of interest (ROIs) and calculating fluorescence changes over time. A detailed protocol is available in supporting information.

Zebrafish *chd1l* knock-out experiments

Zebrafish (*Danio rerio*) were raised and maintained as described in⁶³. The wild type AB strain, the *chd1lsa14029* (TL) mutant line (#15474), carrying the mutation C>T at the genomic location Chr6:36844273 (GRCz11), the Tg(olig2:EGFP)vu12 (AB) (#15211) line were obtained from the European Zebrafish Resource Center. All fish lines reproduce normally, no skewed sex ratio was observed and *chd1l* homozygote mutants were recovered in expected Mendelian ratio. Genotyping of the *chd1l* sa14029 mutant line was performed by a PCR following a restriction digestion. Wholemount immunostaining involved specific protocols for fixation, permeabilization, blocking, and antibody incubation, followed by imaging using MacroFluo ORCA Flash (Leica) system. Additionally, imaging procedures for the oligodendrocyte lineage and Sudan Black B staining were performed on fixed larvae to visualize cellular structures and assess specific cellular populations. For the statistical analysis, at least 3 replicates for each genotype were analyzed and the number of larvae/replicates is indicated. A detailed protocol is available in supporting information.

4. I am curious if the gender and age of PPMS patients influence the methylation status of the 1q21.1 locus.

Response to Comment #4:

All our analysis models include the age and sex as covariates to account for any potential effects on a locus-specific level, as stated in the Methods. In the validation cohort 2, subjects were selected to obtain RRMS and PPMS groups that are matched for age and sex. To address this comment, we have now performed linear regression analysis for all CpGs in the DMR and we did not find any significant association with age, nor did it indicate a notable association with sex in either cohort. Moreover, based on Gatev *et al.*, none of our probes are known as sex-associated CpGs¹⁴. Lack of association with age and sex could be confirmed using large cohorts, e.g., $r^2_{\text{Age-CpG}} = 0.003 - 0.007$ in whole blood ($n > 900$) and cortex ($n > 250$), from the EWAS Database (<https://ngdc.cncb.ac.cn/ewas/datahub>). Therefore, we believe it is unlikely that sex and age in PPMS significantly influence the methylation status of this locus.

5. Did the author also see any changes in the methylation status of the neighbouring TAR region in PPMS patients in addition to the critical region of 1q21.1?

Response to Comment #5:

We thank the Reviewer for this suggestion. In the TAR region, 216/235 CpGs met the quality control and filtering criteria in our 450K data. However, none of 216 TAR CpGs displayed statistically significant differences following adjustment for multiple testing, as depicted below for a fraction of CpGs that displayed nominal significance:

6. Please discuss what factors are likely to contribute to the increased methylation status of the 1q21.1 region in PPMS patients.

Response to Comment #6:

Our data collectively support the notion that genetic variants affect methylation of identified DMR in the 1q21.1 locus. Although, the relationship between SNPs and DNA methylation is complex, the effects of SNPs on proximal DNA methylation can vary depending on the specific genomic context and the presence of other genetic and epigenetic factors. The potential mechanisms mediating genetic control of methylation include direct disruption of the CpG¹⁵, alteration of TF binding sites and/or the three-dimensional chromatin structure or indirect influence on DNA accessibility or methylation machinery by acting on histone posttranslational modifications and non-coding RNAs¹⁶. These can influence the recruitment of DNA methyltransferases or demethylases to specific genomic regions¹⁷.

As a part of methods development and QC, we conducted a thorough examination to determine whether any SNPs in the DMR sequences that associate with high versus low risk of developing MS directly affect CpG sites (as described in the detailed supplementary methods (Supplementary Information): “We have done extensive SNP and sequence analyses to assure that the methylation measurements of the majority of the DMR CpGs are not a result of the technical measurement artefacts driven by a potential effect of SNPs on the CpGs^{12,13}, pyrosequencing or 450K assays.”). This analysis revealed that the CpG sites within the identified DMR in 1q21.1 remained unaffected by SNPs. Moreover, publicly available Hi-C data in cell lines suggest distal and proximal chromatin looping within the extended region, the strongest signal linking the DMR and the region that harbors the identified PPMS-associated SNPs (and encompasses *PRKAB2*, *FMO5*, *CHD1L*, *BCL9*, *ACP6* and *GJA5* genes) (Supplementary Fig. 1). This suggests that the genetic variation associated with PPMS risk might functionally interact with the DMR. In turn, findings from our *in vitro* methylation assay and CRISPR/dCas9 experiments (Fig. 4) indicate that the DMR exerts regulatory properties with methylation-sensitive promoter and enhancer potential, likely regulating *PRKAB2* and *CHD1L* genes expression. The presence of TF binding sites for several CNS-related functions (see response to **Comment #1**), further supports this transcriptional regulation to be cell type-specific.

This has now been clarified in the revised **Discussion** (page 14), which reads as follows:

Among the potential mechanisms suggested to mediate genetic control of methylation, SNPs can influence DNA methylation by directly disrupting CpG sites, altering transcription factor binding sites and/or influencing the three-dimensional chromatin structure⁴⁰⁻⁴². Our annotation of the 1q21.1 DMR showed that the CpG sites were not affected by local SNPs and that chromatin looping exists between the DMR and the region harboring the risk SNPs. Yet, whether such potential physical interaction directly influences the DMR accessibility to epigenetic modifiers such as methylating enzymes or whether the genetic control engages other mechanisms remains to be elucidated.

Reviewer #2 (Remarks to the Author):

This study aimed to identify differentially methylated regions associated with primary progressive MS. The study is warranted based on the current lack of knowledge of disease pathology underpinning progressive MS and the ultimate need for better diagnostics and treatments to prevent this eventual outcome in many MS patients.

The research design although quite complex incorporates multiple independent comparative cohorts (RRMS, SPMS, PPMS and HC), multiple cell types (blood, neurologological) and multiple layers of omic data (genotype, methylation, transcript levels).

Based on the initial EWAS a hypomethylated region on chr 1q21 was convincingly shown to be associated with PPMS. This locus was shown to be under genetic control using blood and neuro meQTL analysis.

Since the meQTL is intergenic the investigators rightly tested for trans-acting effects on gene expression in neuron-like cells incorporating gene editing technology. This nice approach highlighted the wide-spread regulatory effects of the meQTL in these cells. Finally, a bioinformatic-based gene network analysis using publically-available brain data and data from MS patients specifically implicated several genes (and modules) involved in PPMS. Overall, this is a comprehensive study incorporating multiple layers of evidence to causally implicate several genes in neuronal pathology of PPMS. Despite the complex design (workflow) the manuscript is well constructed and written and the conclusions are supported by the results. Overall, this an important and novel piece of research.

Response:

We thank the Reviewer for a thorough and positive review of our manuscript. We hope that our revisions address all suggestions and concerns.

1. One area I think needs addressing is the following. The statistical aspects of the several results sections are very probability (p-value) focused. Whilst this is of course important for interpretation I would like to see some measures of effect size and direction mentioned in the text ie. to accompany the p-values. For example, what is the delta-beta (range, average, max) of the hypomethylation locus on 1q21?

Response to Comment #1:

We thank the Reviewer for this insightful comment. We agree with the biological relevance of the effect size and have now added details in the **Results** section of the revised manuscript:

The DMR, located in an intergenic CG-rich region on chromosome 1 (q21.1): 146549909–146551201 (GRCh37/hg19), consists of 8 consecutive CpG probes displaying hypermethylation in PPMS patients compared to RRMS, SPMS as well as HC, **i.e. with a mean [min-max] $\Delta\beta$ of 0.24 [0.02-0.42], 0.23 [0.03-0.40] and 0.24 [0.02-0.43], respectively** (Fig. 2b).

The two strongest SNPs, rs1969869 and rs4950357, displayed positive association (**β estimates ranging from 1.20 to 1.39**) between the minor allele and CpG methylation levels (Fig. 3a, Supplementary Data 4).

2. The meQTL is under genetic control but what is the strength (heritability) of this? Can you quantify. What are the odds ratios for the meQTL SNPs in the large PPMS cohort association study. All these effects are important to help gauge the relative importance of the finding in the complex context of PPMS pathology.

Response to Comment #2:

We appreciate Reviewer's point of view. The strength of the genetic effect on PPMS risk are listed as ORs in the original **Table 1** and all measures pertaining to the genetic, meQTL and eQTL effects are presented in **Supplementary Data 6, 7 and 8**. We have now further highlighted the effect sizes of the genetic effect on PPMS risk, DMR methylation and genes expression in the **Results** text (page 6 and 7) as well, as follows:

All the protective variants were found to be associated with lower methylation (**β estimates ranging from -0.08 to -0.34**) at 1q21.1 DMR, while 7/8 risk variants for PPMS were associated with higher levels of

methylation (β estimates ranging from 0.09 to 0.73) at 1q21.1 DMR in the brain (Supplementary Data 7).

Five SNPs, tagging two LD blocks, associated with an increased risk of PPMS (OR ranging from 1.48 to 1.84) remained significant after correction for multiple testing accounting for LD blocks ($p < 7.2 \times 10^{-4}$) (Table 1).

Annotation to brain-meQTL data confirmed that risk variants, such as LD block 5 and 6, also lead to higher methylation (β estimates ranging from 0.09 to 0.19, Fig. 3d, Supplementary Data 7).

As exemplified in Figure 3d, variants predisposing for higher DMR methylation and PPMS risk (LD blocks 5 and 6) associated with lower *CHD1L* (normalized enrichment score NES < -0.5) and *PRKAB2* expression (NES < -0.3) while variants conferring low DMR methylation and PPMS protection (LD blocks 3 and 4) associated with higher *CHD1L* and *FMO5* expression (NES > 0.5) and lower *PRKAB2* (NES < -0.3) transcript levels (Supplementary Data 7).

Revised Legend of Fig. 3d: The data shown in this panel are available in Supplementary Data 7 (meQTL, eQTL) and 8 (genetic analysis).

Reviewer #3 (Remarks to the Author):

In this manuscript, the authors explore the molecular mechanism of PPMS through multiple cohorts, cross-tissue, and multilayered data. The results indicated that hypermethylation at 1q21.1 was associated with low expression of CHD1L and PRKAB2 and high risk of PPMS. The manuscript has a large amount of data, complicated statistical work and novel results. However, the manuscript is poorly read, with errors in numerical details and a lack of depth in the discussion of the novel results. Careful adjustment of this manuscript is recommended.

Response:

We thank the Reviewer for the thorough scrutiny of our manuscript and suggestions that have improved the clarity and interpretations of our findings. Based on your comments, and comments from other reviewers, we have revised the manuscript and discussed the existing and new data in more depth.

1. Personally, I don't think this title can cover all the results of this paper.

Response to Comment #1:

We fully agree with the Reviewer. Since our original data particularly highlighted a link between *CHD1L* and PPMS, we focused during the revision on investigating if *CHD1L* impacts CNS. We conducted extensive experiments using two models, i.e., *in vitro* human iPSC-derived neuronal *CHD1L* knock-down model and *in vivo chd1l* knock-out model in zebrafish. Our new data demonstrate a role of *CHD1L* in CNS development and neuronal functioning in particular (*for details, please see response to Comment #3 from Reviewer #1*). To emphasize these findings, while keeping within Nature Communication's limit of 15 words, we have revised the title that now reads as follows (14 words):

A genetic-epigenetic interplay at 1q21.1 locus underlies *CHD1L*-mediated vulnerability to Primary Progressive Multiple Sclerosis

2. Please check whether multiple sclerosis is a neurodegenerative disease. I couldn't find any definition of neurodegenerative disease in the first reference (Kutzelnigg, A. et al Cortical demyelination and diffuse white matter injury in multiple sclerosis. Brain).

Response to Comment #2:

Although MS is traditionally considered as a chronic inflammatory and demyelinating diseases of the CNS, neurodegeneration is the primary driver of disease manifestation and progression. Notably, while clinical progression is a late phenomenon, it is now well-established that neurodegeneration can occur early in disease development as well as in absence of focal lesions. Historically, there has been an emphasis on focal lesions, as their features such a blood brain barrier breakdown, infiltration of activated peripheral immune cells in the white matter and demyelination, can readily be detected with conventional MRI scanners¹⁸. Such lesions correlate with relapses but not with progression of disability, they are much rarer in later progressive disease, which is instead dominated by chronic active lesions displaying reactive astrogliosis and activated microglia¹⁹. Cortical lesions also occur early in the disease and become more prevalent in progressive stages. In addition to the focal lesions, macroscopically normal-appearing brain tissue is also affected by neuro-axonal damage and diffuse inflammation^{20,21}. Numerous studies have suggested that once a clinical threshold of irreversible disability is attained, disability progression becomes independent of prior clinical events such as early focal inflammation mirroring relapses. This further prompted the definition of 'progression independent of relapse activity' (PIRA), which is frequently observed in patients and appears to account for most of disability accumulation²²⁻²⁴. This aligns with our recent heritability enrichment analysis confirming the immune nature of susceptibility to develop MS and the CNS determinant of disease severity and progression, implicating mechanisms related to neuronal resilience and regeneration²⁵.

The reference cited in the original manuscript does not reflect this notion. To clarify the role of neurodegenerative mechanisms to the readers outside the MS field, we have now modified the sentence in the introduction and included a more pertinent reference, as indicated below:

Introduction

Multiple Sclerosis (MS) is an inflammatory and neurodegenerative disease of the central nervous system (CNS) affecting young adults and leading to unpredictable and progressive physical and cognitive impairments. The pathological hallmarks of MS are represented by focal plaques of primary demyelination and diffuse neurodegeneration in the grey and white matter of the brain and spinal cord¹.

1- Stadelmann, C. Multiple sclerosis as a neurodegenerative disease: pathology, mechanisms and therapeutic implications. *Current opinion in neurology* **24**, 224-229 (2011).

3. The authors used at least 5 cohorts to investigate the methylation of the PPMS-associated 1q21.1 locus, but didn't describe the details and reasons for using of all the cohorts. They were messy and wasn't closely related. Why didn't use cohort 1 and cohort 2 but cohort 3 to proceed association analysis? Is the brain tissue used to locus-specific meQTL and correlation network analysis from the same cohort? In the second part of result, the authors mentioned 'only eight out of the 19 meQTL-SNPs identified in cohort 1 could be assessed in this cohort (cohort 2)', would some important information be left out? Was it limited by technology or the quality of blood samples or something else?

Response to Comment #3:

We thank the Reviewer for pointing the lack of clarity. The subject selection as well as methods to measure DNA methylation, genotype and expression, varied depending on the original purpose of the study cohort. For example, while cohort 1 was used for unbiased genome-wide analysis (thus, methylation and genotypes were measured using genome-wide arrays), cohort 2 was specifically selected to replicate and validate the findings from cohort 1 in a locus-specific manner. Thus, in cohort 2, independent and larger number of PPMS subjects was selected together with sex- and age-matched RRMS. Methylation and genotype were analyzed with targeted methods, i.e., pyrosequencing was used to measure methylation of the selected significant CpGs from cohort 1 and only genotypes from the locus were used in analysis. Due to differences in methodologies, not all SNPs have been profiled in cohort 1 and cohort 2 (although their information is captured by other SNPs that are in high linkage disequilibrium), which answers the second part of Reviewer's comment.

While cohort 1 and 2 had sufficient size to establish effect of SNPs on methylation (meQTLs) and associated methylation differences between PPMS and other clinical groups, establishing genetic association with the risk of developing PPMS requires much larger cohorts given the complex multifactorial nature of the disease. For this reason, to conduct genetic association analysis, we gathered as many available genotyped PPMS cases for cohort 3, with the majority of subjects from cohort 1 and 2 being also included in this cohort.

Cohort 1-3 comprises clinical data and blood samples retrieved from living subjects, which is why we could not use any of these subjects for the brain locus-specific meQTL and correlation network analysis. These investigations necessitated molecular profiling of *post-mortem* brain tissue samples which was conducted in independent cohorts (*detailed further below*).

A detailed description of the cohorts has now been added in the revised manuscript (**Supplementary Data 1**).

Cohort 1: comprised MS (represented by RRMS, SPMS and PPMS) and sex- and age-matched healthy controls. This cohort was utilized for an unbiased genome-wide analysis of methylation differences between the clinical groups and subsequent meQTL analysis. Methylation was measured using Illumina Infinium HumanMethylation450 BeadChip and samples were genotyped with Illumina arrays.

Cohort 2: comprised sex- and age-matched PPMS and RRMS. This cohort was utilized for an independent replication cohort for locus-specific methylation differences and meQTLs identified in cohort 1. Methylation of 1q21.1 CpGs was measured using pyrosequencing and samples were genotyped with Illumina OmniExpress chip. The difference of genotyping methodologies explains the imperfect overlap of genotyped SNPs between cohort 1 and cohort 2.

Cohort 3: comprised PPMS and bout-onset MS (BOMS, i.e., RRMS and SPMS), selected from Sweden and Italy to increase the statistical power. This cohort was utilized to investigate the association of the SNPs influencing 1q21.1 methylation (identified in cohort 1 and 2) and the susceptibility to develop

PPMS. High-quality imputed genetic variants from the extended 1q21.1 region were used. Most of the individuals from cohort 1 and 2 are included in cohort 3.

The brain locus-specific meQTL and correlation network analysis, necessitated *post-mortem* brain tissue samples that were not available from cohort 1-3 (comprising blood samples for genotyping and methylation from living subjects):

- As detailed in the results section (page 5), specifically in “*Genetic control of 1q21.1 methylation in the blood and CNS*”, our brain locus meQTL analysis used publicly available data from brain tissue of 543 individuals from xQTL serve database: “Our results, implying a link between genetic variation and blood DNA methylation at the 1q21.1 region, a locus reported to be crucial for brain size¹⁶⁻¹⁸ and neuropsychiatric disorders¹⁹⁻²¹, posed the question whether such effect might occur in the CNS as well. To address this, we examined putative genetic influences on the methylation at 1q21.1 DMR in the brain using publicly available meQTL data from brain tissue of 543 individuals¹⁵.”
- For the correlation network analysis, we utilized bulk RNA-sequencing-derived gene expression matrices from brain tissue samples of progressive MS patients and non-neurological controls, highlighted in the results section (page 10), specifically “To further delineate the putative relevance of the genes included in the 1q21.1 locus in PPMS brain pathology, we constructed an unbiased correlation network analysis using RNA-sequencing-derived gene expression matrix from brain tissue samples of progressive MS patients (n = 12) and non-neurological controls (n = 10)⁹ (Fig. 5).”
- To validate our network findings, we exploited a large GTEx dataset from healthy individuals and analyzed postmortem brain snRNA-seq data from MS patients and could confirm that the *CHD1L* co-expressed network is disease-dependent and likely more restricted to neurons (all datasets are listed in Supplementary Data 10) and validation cohorts referenced in: “*CHD1L gene network is enriched in PPMS brain tissue*”.

All the information sourced from various cohorts has been compiled in Methods (and Supplementary Data) under “Cohorts” and is detailed in Supplementary Data 1 & 11 and Figure 1.

4. The authors didn't discuss why locus-specific demethylation at CpGs of the identified DMR at the 1q21.1 DMR did not affect any significant gene expression changes in HEK293T cells? And why 'the expression of FMO5 and other proximal transcripts, PDIA3P1 and PFN1P5, did not vary significantly'.

Response to Comment #4:

We thank the Reviewer for highlighting the lack of clarity with respect to the purpose of HEK293T experiments in the study. Since HEK293T cells are easy to transfect, even with large plasmid constructs such as ours, they were (as commonly done) utilized solely for the purpose of screening and selecting gRNAs that induced efficient demethylation at the locus of interest. A similar gRNA screen would not be feasible in more relevant - but hard-to-transfect cells - or it would require substantial resources for viral transduction-based gRNA screen. HEK293T is a noncancerous human embryonic kidney epithelial cell line²⁶, thus not suitable for investigating cellular or molecular processes relevant to MS. It is highly likely that these cells do not capture either the epigenetic landscape and transcriptional regulation that is relevant in the context of MS (either immune or brain tissue), which could explain the lack of regulation of gene expression following demethylation.

For these reasons, we opted for SH-SY5Y a human neuron-like cell line, which we consider more functionally relevant for our study. Moreover, SH-SY5Y displays hypermethylation at the 1q21.1 locus (ENCODE Roadmap) mimicking methylation pattern in PPMS patients. These features made SH-SY5Y suitable to address the question relating the methylation status of the 1q21.1 DMR to the expression level of surrounding genes. expression using CRISPR/dCas9-TET1. Indeed, demethylation of the 1q21.1 DMR led to increased expression levels of only *CHD1L* and *PRKAB2*, possibly due to the nature of the regulatory properties of an intergenic regulatory locus allowing differential usage in a cell-type specific manner. Publicly available Hi-C data in cell lines at this locus suggest distal and proximal chromatin looping within the extended region (Supplementary Fig. 1). Such cell type-specific gene regulation is further supported by the differential eQTL effects across various tissues reported by GTEx database. We have now clarified this in the revised manuscript as follows:

Revised **Methods** (page 21): *dCas9-TET1 delivery to HEK293T and SH-SY5Y cells.* To test the efficiency of epigenome editing, we exploited HEK293T (female human embryonic kidney epithelial cell line⁵⁵) ease of transfection and performed gRNAs screen. Different gRNAs were transfected either individually or in combination based on the target site, using Lipofectamine 3000 (Invitrogen).

Revised **Discussion** (page 16): *Yet, CHD1L is likely to influence the functionality of other cell types involved in the CNS development, such as oligodendrocyte lineage as suggest by results in the zebrafish, and one cannot exclude additional influences of other genes, such as PRKAB2, from the 1q21.1 locus⁴⁵.*

5. Can the authors explain the correlation among the expression level of PRKAB2, the methylation level and the PPMS risk? (line 197-200)

Response to Comment #5:

This is indeed a relevant observation, i.e., that there is no consistent association between the expression level of *PRKAB2*, the methylation level and the PPMS risk. While the variants predisposing for PPMS risk (high DMR methylation) versus protective (low DMR methylation) have coherent opposite effect on *CHD1L* expression, it seems that both risk and protective SNPs associate with lower *PRKAB2* expression. While the reasons behind this association are elusive, this, together with other compelling evidence (all listed below) further directed our functional experiments during revision on *CHD1L*:

- Our reporter and CRISPR experiments demonstrate that hypermethylation at the 1q21.1 DMR represses gene expression (Fig. 4) and concordant meQTL/eQTL effect was observed for *CHD1L* within the CNS tissue, as opposed to *PRKAB2* (Fig. 3d).
- The gene ontology analysis of the *CHD1L* network in the PPMS postmortem brain revealed functional annotations related to neuronal and neurodegenerative processes, those were not detected for *PRKAB2* (Fig. 5c).
- Notably, we detected a significant overlap of the *CHD1L* gene network (but not of the *PRKAB2* and *FMO5* networks), with the gene signature of CUX2⁺ excitatory neurons, found by Schirmer *et al.* to be specifically affected in patients with MS¹², with the closest interactors of *CHD1L* (but not *PRKAB2* and *FMO5*) also replicated in CUX2⁺ excitatory neurons.

Given these lines of evidence emphasizing the significance of *CHD1L* in the context of PPMS, we conducted functional experiments using *in vitro* using iPSC-derived *CHD1L* knock-down neuronal model and *in vivo chd1l* knock-out model in zebrafish. We demonstrated that loss of *CHD1L* influences the development and function of neurons (*for details of conducted experiments, results and the according changes in the revised manuscript, please see responses to Comment #3 from Reviewer #1*).

This has now been clarified in the revised **Results** (page 11) and reads as follows: *Thus, we detected a significant overlap between PPMS-associated CHD1L gene network and the gene signature of neurons previously reported to be specifically affected in MS²⁴, which, together with the concordant meQTL and eQTL effects of PPMS-associated variants for this gene, implicate CHD1L gene in neuronal vulnerability in PPMS.*

Revision in **Discussion** (page 16) reads as follows: *Yet, CHD1L is likely to influence the functionality of other cell types involved in the CNS development, such as oligodendrocyte lineage as suggest by results in the zebrafish, and one cannot exclude additional influences of other genes, such as PRKAB2, from the 1q21.1 locus⁴⁵.*

6. Please add the legends of Figure3D. What does the variant ID represent? It is difficult to read.

Response to Comment #6:

We thank the Reviewer for pointing out the lack of clarity. The “variant ID” represented the LD blocks (also listed in Table 1) predisposing/protecting for PPMS identified in the meta-analysis of the Swedish and Italian cohort. We have now clarified this in the figure, figure legend and corresponding text, as follows:

Revised Fig. 3 panel legend: d. Association between genetic variation, DNA methylation (using xQTL serve platform¹¹) and gene expression (using GTEx database) in the extended locus for SNPs tagging four variants (LD blocks) displaying significant association with PPMS in the meta-analysis of Swedish (SWE) and Italian (ITA) cohorts. Protective and risk genetic effects are depicted in green and orange, respectively. Green and orange colors reflect protective and risk effect on PPMS risk conferred by the genetic variants. Blue and red colors represent negative and positive effect of the minor allele, respectively, on methylation or expression. The data shown in this panel are available in Supplementary Data 7 (meQTL, eQTL) and 8 (genetic analysis).

Corresponding text in the revised Results (page 7): As exemplified in Figure 3d, variants predisposing for higher

DMR methylation and PPMS risk (LD blocks 5 and 6) associated with lower *CHD1L* (normalized enrichment score NES < -0.5) and *PRKAB2* expression (NES < -0.3) while variants conferring low DMR methylation and PPMS protection (LD blocks 3 and 4) associated with higher *CHD1L* and *FMO5* expression (NES > 0.5) and lower *PRKAB2* (NES < -0.3) transcript levels (Supplementary Data 7).

7. Line 249, I couldn't find the related information in the figure 3C. Please label the correct figure.

Response to Comment #7:

Thank you for noticing this error. The correct annotation is Fig. 3d and not 3c and has been amended accordingly in the revised text: These include the three coding genes found regulated by the same meQTL-SNPs in the normal brain (Fig. 3d).

8. In the fifth section of result, the authors indicated 'but *CHD1L*, *FMO5* and *PRKAB2* genes belonged to significantly co-expressed genes networks only in the cortex and tibial nerve', but the typical pathology of MS is focal plaques of primary demyelination. However, the authors didn't discuss the different location of the pathology and genes co-expression in the part of discussion.

Response to Comment #8:

The genes co-expression analysis performed using the large GTEx dataset suggested that the gene networks identified in the PPMS bulk brain might be disease-dependent and likely neuron-related, which we next ascertained using higher resolution (single nuclei) RNA-seq data from MS postmortem brain samples. We have clarified the role of neuro-axonal damage in MS (see response to **Comment #1 from Reviewer #3**) and the grounds for our focus on the role of *CHD1L* in neuronal cells (response to **Comment #3 from Reviewer #1 and Comments 5 from Reviewer #3**). We have also included additional functional experiments that demonstrate a role of *CHD1L* in neuronal development and function (response to **Comment #3 from Reviewer #1**). The accompanying changes and additions have been done in the revised **Methods**, **Results** and **Discussion** sections and new **Fig. 6**, **Supplementary Fig. 5** included.

9. To validate the bioinformatics data, the authors used SH-SY5Y cell line and HEK293T cell line, which are mostly used for modeling neurodegenerative diseases. Are they the recognized cell model for MS?

Response to Comment #9:

We appreciate the Reviewer's comments and would like to highlight the fact that the use of these cell lines was not to "model MS", but rather to identify a link between methylation changes at the 1q21.1 locus and transcript levels of *CHDL1* and other genes within the locus. The purpose of using HEK293T (gRNA screen, purely technical) and SH-SY5Y (neuronal-like cell line with hypermethylation of 1q21.1 mimicking PPMS, used to molecularly link methylation with gene expression) has been described in detail earlier (response to **Comment #4 from the Reviewer #3**).

MS is characterized by an interplay of intertwined processes including immune dysregulation, neuroinflammation, demyelination and neurodegeneration, posing challenges for recapitulating such complexity in cellular models²⁷. We and others commonly exploit different *in vitro* and *in vivo* models that can be applied in a context of a particular pathology of interest. As such, in this study we used HEK293T and SH-SY5Y cell lines, CNS tissues and isolated CNS cell types (in-house and public), *in vitro* human iPSC-neurons and *in vivo* zebrafish models, relying on cumulative evidence from these different models in supporting a role of 1q21.1 and more particularly *CHD1L* in PPMS.

The functional experiments in human iPSC-derived neurons and *in vivo* zebrafish model that demonstrate a role of *CHD1L* in neuronal development and function (*response to Comment #3 from Reviewer #1*). The accompanying changes and additions have been done in the revised **Methods**, **Results** and **Discussion** sections and new **Fig. 6**, **Supplementary Fig. 5** included.

10. Please confirm the details of the data in the article. For example, 140 HC in cohort 1 is shown in Figure 1. According to the description in the manuscript, it seems that the number of HC should be 139.

Response to Comment #10:

Thank you for noticing this error that has now been corrected in the revised manuscript. Figure 1 has also been further amended to include functional *CHD1L* experiments done during revision:

Figure 1

References:

- 1 Gao, Z. *et al.* Neurod1 is essential for the survival and maturation of adult-born neurons. *Nature neuroscience* **12**, 1090-1092 (2009).
- 2 Rose, M. F. *et al.* Math1 is essential for the development of hindbrain neurons critical for perinatal breathing. *Neuron* **64**, 341-354 (2009).
- 3 Chanda, S. *et al.* Generation of induced neuronal cells by the single reprogramming factor ASCL1. *Stem cell reports* **3**, 282-296 (2014).
- 4 Aprato, J. *et al.* Myrf guides target gene selection of transcription factor Sox10 during oligodendroglial development. *Nucleic acids research* **48**, 1254-1270 (2020).
- 5 Ueno, T. *et al.* The identification of transcriptional targets of Ascl1 in oligodendrocyte development. *Glia* **60**, 1495-1505 (2012).
- 6 Chaboub, L. S. & Deneen, B. in *Seminars in pediatric neurology*. 230-235 (Elsevier).
- 7 Trubetskov, V. *et al.* Mapping genomic loci implicates genes and synaptic biology in schizophrenia. *Nature* **604**, 502-508 (2022).
- 8 Chapman, G. *et al.* Using induced pluripotent stem cells to investigate human neuronal phenotypes in 1q21. 1 deletion and duplication syndrome. *Molecular Psychiatry* **27**, 819-830 (2022).
- 9 Nomura, Y., Nomura, J., Nishikawa, T. & Takumi, T. Reciprocal differentiation via GABAergic components and ASD-related phenotypes in hES with 1q21. 1 CNV. *bioRxiv*, 2021.2009.2013.460033 (2021).
- 10 Søndersby, I. E. *et al.* 1q21. 1 distal copy number variants are associated with cerebral and cognitive alterations in humans. *Translational psychiatry* **11**, 182 (2021).
- 11 Ng, B. *et al.* An xQTL map integrates the genetic architecture of the human brain's transcriptome and epigenome. *Nat Neurosci* **20**, 1418-1426, doi:10.1038/nn.4632 (2017).
- 12 Schirmer, L. *et al.* Neuronal vulnerability and multilineage diversity in multiple sclerosis. **573**, 75-82 (2019).
- 13 Dou, D. *et al.* CHD1L promotes neuronal differentiation in human embryonic stem cells by upregulating PAX6. *Stem cells and development* **26**, 1626-1636 (2017).
- 14 Gatev, E. *et al.* Autosomal sex-associated co-methylated regions predict biological sex from DNA methylation. *Nucleic Acids Research* **49**, 9097-9116 (2021).
- 15 Bell, J. T. *et al.* DNA methylation patterns associate with genetic and gene expression variation in HapMap cell lines. *Genome biology* **12**, 1-13 (2011).
- 16 Maurano, M. T. *et al.* Role of DNA methylation in modulating transcription factor occupancy. *Cell reports* **12**, 1184-1195 (2015).
- 17 Degner, J. F. *et al.* DNase I sensitivity QTLs are a major determinant of human expression variation. *Nature* **482**, 390-394 (2012).
- 18 Thompson, A. J. *et al.* Diagnosis of multiple sclerosis: 2017 revisions of the McDonald criteria. **17**, 162-173 (2018).
- 19 Luchetti, S. *et al.* Progressive multiple sclerosis patients show substantial lesion activity that correlates with clinical disease severity and sex: a retrospective autopsy cohort analysis. **135**, 511-528 (2018).
- 20 Kutzelnigg, A. *et al.* Cortical demyelination and diffuse white matter injury in multiple sclerosis. **128**, 2705-2712 (2005).
- 21 Evangelou, N., DeLuca, G., Owens, T. & Esiri, M. J. B. Pathological study of spinal cord atrophy in multiple sclerosis suggests limited role of local lesions. **128**, 29-34 (2005).
- 22 Kappos, L. *et al.* Greater sensitivity to multiple sclerosis disability worsening and progression events using a roving versus a fixed reference value in a prospective cohort study. **24**, 963-973 (2018).

- 23 Kappos, L. *et al.* Contribution of relapse-independent progression vs relapse-associated worsening to overall confirmed disability accumulation in typical relapsing multiple sclerosis in a pooled analysis of 2 randomized clinical trials. **77**, 1132-1140 (2020).
- 24 Cagol, A. *et al.* Association of brain atrophy with disease progression independent of relapse activity in patients with relapsing multiple sclerosis. **79**, 682-692 (2022).
- 25 %J Nature, M. C. H. A. J. I. B. Y. L. S. M. L. S. A. V. P. S. K. Locus for severity implicates CNS resilience in progression of multiple sclerosis. **619**, 323-331 (2023).
- 26 Graham, F. L., Smiley, J., Russell, W. & Nairn, R. Characteristics of a human cell line transformed by DNA from human adenovirus type 5. *Journal of general virology* **36**, 59-72 (1977).
- 27 Julian, K. & Imitola, J. in *Current Progress in iPSC Disease Modeling* 31-43 (Elsevier, 2022).

REVIEWER COMMENTS

Reviewer #1 (Remarks to the Author):

The authors have made a substantial effort revising the manuscript and revised manuscript is greatly improved.

I have a minor comment on Fig6, where the authors show a reduction in spontaneous calcium activity post knock down of CHDIL in iPSC derived neural stem cells. While cortical markers do not vary following neuronal differentiation, could the difference in spontaneous calcium activity be attributed to the level neuronal maturity between groups? This aspect could be investigated by examining the level of mature neuronal markers such as MAP2 and the immature marker DCX. Furthermore, I recommend including representative calcium traces among experimental groups in the figure.

Reviewer #3 (Remarks to the Author):

I have thoroughly and critically read the response to reviewers comments. It is very evident that the authors have provided detailed responses with appropriate supporting results and amendments to the manuscript. Overall, I am satisfied with the revisions and am happy to recommend acceptance of the manuscript.

Point-by-point response to reviewers' comments

Reviewer #1 (Remarks to the Author):

The authors have made a substantial effort revising the manuscript and revised manuscript is greatly improved.

Response:

We would like to thank the Reviewer for the constructive comments and suggestions that have significantly improved our manuscript.

I have a minor comment on Fig6, where the authors show a reduction in spontaneous calcium activity post knock down of *CHD1L* in iPSC derived neural stem cells. While cortical markers do not vary following neuronal differentiation, could the difference in spontaneous calcium activity be attributed to the level neuronal maturity between groups? This aspect could be investigated by examining the level of mature neuronal markers such as MAP2 and the immature marker DCX. Furthermore, I recommend including representative calcium traces among experimental groups in the figure.

Response to Comment:

We thank the reviewer for these suggestions. We observed similar staining for MAP2 in NT and *CHD1L* KD cultures, as depicted below. We have also now performed qPCR on neuronal RNA from NT and *CHD1L* KD samples and found no change in the expression of immature neuronal marker *DCX*. The new data is now presented in Figure 6 which also now includes representative calcium traces from 3 ROIs reflecting differences captured between the experimental groups (see below). The corresponding text in the revised **Result** section reads as follows:

Results (p. 12): The successful knock-down of *CHD1L* during differentiation did not affect the expression of intermediate progenitor and migration markers (*EOMES*, *NCAM1*), cortical layer markers (*CTIP2*, *SATB2*) or immature neuronal marker (*DCX*) (Fig. 6g). Instead, *CHD1L*-deficient neurons presented branching abnormalities reflected by smaller MAP2-positive neuritic protrusions, while the synaptic density seemed unaffected (Fig. 6h). *CHD1L*-knockdown also resulted in neuronal functional impairment detectable as reduced calcium intensity signal, overall suggesting weaker electrical activity in neurons with lower levels of *CHD1L* (Fig. 6i-j).

Revised Fig. 6:

Figure 6

Revised Fig. 6 panel legend:

Figure 6. Functional impact of low *CHD1L* expression in neurons and oligodendrocytes. **a.** Schematic of the experimental design in the zebrafish. **b.** Dorsal view of control and *chd1l*^{+/-} larvae at 4 dpf stained with acetylated tubulin. Barplot of the inter-tecta axonal tract projections count after acetylated tubulin staining of 3 dpf control and *chd1l*^{+/-} larvae (mean ± SEM for n = 3 replicate per genotype with 30 larvae/replicate). **c.** Dorsal view of *Tg(olig2:EGFP); chd1l*^{+/+} and *Tg(olig2:EGFP); chd1l*^{+/-} at 3 dpf with the hindbrain used for olig2-positive cells count outlined in yellow. Barplot of the number of olig2-positive cells per μm² in the hindbrain of *Tg(olig2:EGFP); chd1l*^{+/+} and *Tg(olig2:EGFP); chd1l*^{+/-} larvae at 3 dpf (mean ± SEM of n = 3 per genotype and 12-20 larvae/replicate). **d.** Lateral view of the spinal cord of *Tg(olig2:EGFP); chd1l*^{+/+} and *Tg(olig2:EGFP); chd1l*^{+/-} larvae at 3 dpf. Barplot of the number of olig2-positive cells migrating dorsally from the spinal cord of *Tg(olig2:EGFP); chd1l*^{+/+} and *Tg(olig2:EGFP); chd1l*^{+/-} larvae at 3 dpf (mean ± SEM of n = 3 replicate per genotype and 15-20 larvae/replicate). **e.** Schematics showing *in vitro* differentiation of iPSCs into forebrain neurons. **f.** Representative confocal images of markers of neuroectodermal NPCs (PAX6, SOX1) and differentiated neurons (MAP2, SYN1). **g.** qPCR data of *CHD1L*, *EOMES*, *SATB2*, *CTIP2* and *DCX* transcripts in non-targeting (NT) and *CHD1L* knock-down (KD) neurons. Data points correspond to experimental duplicates in three independent subject lines (highlighted in different colors). **h.** Quantification of neurite length (μm) and synaptic density in NT and KD neurons. **i.** Quantification of background-corrected changes in calcium-sensitive dye fluorescent intensity in NT and *CHD1L* KD neurons. A representative confocal image is shown. Data points correspond to averaged active regions (ROIs, 10 cells) per field of view. Five to six fields of view were included for each subject line and condition. **j. Representative calcium imaging traces from NT and *CHD1L* KD neurons (3 ROIs).** Statistical significance was assessed using the Wilcoxon test (b, c.), Student's T-test (d), Mann-Whitney U test (g, h, i) for p-value. A, anterior; P, posterior; dpf, days post-fertilization; NT, non-targeting siRNA; KD, *CHD1L*-targeting siRNA.

REVIEWERS' COMMENTS

Reviewer #1 (Remarks to the Author):

The authors have satisfactorily addressed my comments. I endorse the publication of the manuscript